# Drought and vegetation change in the central Rocky Mountains: Potential climatic mechanisms associated with drought conditions at 4200 cal yr BP.

Vachel A. Carter[1,2], Jacqueline J. Shinker[3], Jonathon Preece[3]

[1]RED Lab, Department of Geography, University of Utah, Salt Lake City, UT, 84112, USA
[2]Department of Botany, Charles University, Prague, 12801, Czech Republic
[3]Department of Geography and Roy J. Shlemon Center for Quaternary Studies, University of Wyoming, Laramie, WY, 82071, USA

*Correspondence to*: Vachel A. Carter (vachel.carter@gmail.com)

**Abstract.** Droughts are a naturally re-occurring phenomena that result in economic and societal losses. Yet, the most historic droughts that occurred in the 1930s and 1950s in the Great Plains and western United States were both shorter in duration, and less severe than mega droughts that have plagued the region in the past. Roughly 4200 years ago, a ~150-year long mega drought occurred in the central Rocky Mountains, as indicated by sedimentary pollen evidence from Long Lake, south-eastern Wyoming. However, pollen evidence does not record the climate mechanisms that caused the drought; they only provide evidence that the drought occurred. A modern climate analogue technique using North American Regional Reanalysis data was applied to the sedimentary pollen data to provide a conceptual framework for exploring possible mechanisms responsible for the observed ecological changes. The modern climate analogues illustrate that persistent warm and dry conditions throughout the growing season were a result of anomalously higher-than-normal geopotential heights centred over the Great Plains, which suppressed moisture transport via the low level jet from the Gulf of Mexico, as well as brought in dry continental air from the interior region of North America. Associated climatic responses are consistent with local and regional proxy data suggesting regional drought conditions ~4200 cal yr BP in the central Rocky Mountains and central Great Plains. Persistent drought-like conditions provide insight as to the mechanisms that facilitated abrupt changes in vegetation composition in the past in our study region of south-eastern Wyoming.

## 1. Introduction

Throughout the 20th century, droughts have caused both economic and societal losses throughout the Great Plains and the western United States (US) (Diaz, 1983; Karl and Koscielny, 1982; Karl, 1983; Woodhouse and Overpeck, 1998). However, these most recent droughts were both shorter in duration, and less severe than mega droughts that have plagued the region in the past (Woodhouse and Overpeck, 1998). Globally averaged land and ocean temperatures have increased in recent decades, which have affected the hydrological cycle and subsequent moisture availability in these regions (IPCC, 2014). Subsequently,

drought vulnerability has increased in both the Great Plains and western US (Garfin et al., 2013). In the Upper Platte River drainage basin ( the region of interest in this study), there has been a 1.44ºC increase in temperature since 1916, leading to the earlier onset of spring snowmelt by approximately 11 days (Shinker et al, 2010). Global climate change is projected to yield continued changes in moisture availability and the frequency or intensity of drought (Breshears et al., 2005; Cayan et al., 2010; Seager et al., 2007; Sterl et al., 2008). Such changes in the hydrological cycle have the potential to influence hydrologic systems, ecosystem processes, and vegetation distribution in the future.

Since 1970, widespread forest mortality on a global-scale has occurred as a result of water or heat stress (Allen et al., 2010). Drought and increased temperatures have resulted in widespread mortality among certain western North American , quaking aspen (*Populus tremuloides* Michx.) communities (Anderegg et al., 2013a; 2013b; Hanna and Kulakowski, 2012; Hogg et al., 2008; Kashian et al., 2007; Rehfeldt et al., 2009; Worrall et al., 2008; 2010; 2013). Quaking aspen is the most widely distributed deciduous tree species in North American, and is considered a keystone species in the Intermountain West region (Kay, 1997; Little, 1971; Perala, 1990). While the recent decline of quaking aspen appears to be influenced by increased temperatures, the relationship between climate variability and quaking aspen mortality remains poorly understood prior to the 20[th] century (Anderegg et al., 2013b; Hanna and Kulakowski, 2012; Worrall et al., 2010). Carter et al. (2013) documented a unique change in vegetation composition ~4000 cal yr BP from sedimentary proxy evidence from Long Lake, south-eastern Wyoming in the Upper Platte River drainage basin. Proxy evidence from Long Lake documents a change from a pine-dominated forest to a mixed-forest with pine and quaking aspen which the authors have coined as the '*Populus* period.' Subsequently, Carter et al. (2017a) investigated the role that climate variability and wildfire activity had on the persistence of quaking aspen during the *Populus* period, and determined that increased temperatures associated with a temporally constrained ~150-year long drought between 4300 and 4100 cal yr BP likely influenced the upslope migration of quaking aspen stands in the Medicine Bow Range of south-eastern Wyoming, while frequent fire activity may have led to the ~500-year persistence of quaking aspen. This ~150-year long drought may be associated with the 4.2 ka 'megadrought' that occurred throughout the Great Plains region (Booth et al., 2005).

While sedimentary proxy data such as pollen and charcoal provide a record of changes in past vegetation and disturbances such as fires or drought, they do not provide a record of the climatic mechanisms that initially caused such extreme events. Therefore, understanding modern climate mechanisms and processes that cause drought conditions, and the sensitivity and range of ecological responses to drought are fundamental to ecosystem management (Mock and Shinker, 2013). Improving our understanding of the climate processes associated with modern drought will provide better insight about past drought variability, and the mechanisms that caused mega droughts evident in paleoecological records. A better understanding about drought and ecological responses, especially in regions with high topographic and climatic variability is important information that can be applied to land managers (Shinker et al., 2006). One tool we use can use to understand the climate processes associated with past drought seen in proxy data is through the use of the modern climate analogue technique (Mock and

Shinker, 2013), which relies on the principle of uniformitarianism and assumes that modern synoptic and dynamic climate processes operated similarly in the past as they do today. The goal of the modern climate analogue technique is to provide examples of how synoptic processes work in order to explain paleoclimatic variations in the past (Mock and Shinker, 2013), and is therefore an effective way to identify climate mechanisms associated with past environmental changes (e.g. as seen in reconstructed sedimentary pollen analyses) (Edwards et al., 2001; Mock and Brunelle-Daines, 1999; Shinker et al., 2006; Shinker, 2014; Carter et al., 2017b). A modern climate analogue is a conceptual model that uses modern extremes (e.g. drought) as analogues of past events (e.g. vegetation disturbance associated with drought in the sedimentary record) as a means to understand palaeoclimate patterns that may have caused extremes in the past (Diaz and Andrews, 1982; Ely, 1997; Edwards et al., 2001; Shinker, 2014). Such conceptual models of dynamic processes can provide examples of modern climatic mechanisms in order to explain historic palaeoclimate variability (Mock and Bartlein, 1995; Mock and Brunelle-Daines, 1999; Mock and Shinker, 2013; Shinker et al. 2006; Shinker, 2014; Carter et al., 2017b).

The modern climate analogue approach has been previously used to understand past synoptic processes and ecological changes recorded in paleoenvironmental data. For example, Mock and Brunelle-Daines (1999) investigated how summer synoptic climatology and external forcing (i.e. Milankovich cycles) impacted effected moisture in the western US during the mid-Holocene (~6000 cal yr BP). Similarly, Edwards et al. (2001) used the modern climate analogue technique to understand how specific atmospheric circulation patterns could have caused surface temperature and effective moisture anomalies during the past 12,000 years in the interior of Alaska. Shinker et al. (2006) also examined the mid-Holocene drought, but focused on the mid-continent of North America to provide potential climate processes and mechanisms associated with low lake levels during the prolonged mid-Holocene drought. By applying the modern climate analogue technique to paleoenvironnmental proxies from the mid-continent, Shinker et al. (2006) found that regional moisture influx and small-scale vertical motions in the atmosphere (i.e. subsidence or uplift) provide better information regarding precipitation than large-scale general circulation alone. The authors' stress that controls of surface and atmospheric processes must also be addressed in paleoclimate reconstructions as these can override the influence of broad circulation patterns (Shinker et al., 2006). Shinker (2014) used the modern climate analogue technique to understand climatic controls on water resources in the headwaters of the Upper Arkansas River basin in west-central Colorado and found that local-scale variations in moisture availability and the absence of uplift mechanisms were key in explaining hydroclimate variability evidenced in paleo lake-level reconstructions in the Upper Arkansas River basin (Shuman et al., 2009; 2010). Finally, Carter et al. (2017b) used modern climate analogues to describe how atmospheric conditions created unique spatial patterns of wildfire activity over the 1200 years in the northern and southern Rocky Mountains, as seen in 37 sedimentary charcoal records.

Here we apply a modern climate analogue approach similar to Shinker et al. (2006; 2014) to a paleoecological reconstruction from Long Lake, Wyoming by using an environment-to-circulation approach (Barry and Carleton, 2001; Mock and Shinker, 2013; Shinker, 2014; Yarnal, 1993; Yarnal et al., 2001). An environment-to-circulation approach considers the surface

conditions based on information gained from the proxy data collected from lake sediments (Mock and Shinker, 2013). In this study, the proxy data were collected from Long Lake, Wyoming (Carter et al., 2013; 2017a). By identifying extremes in the modern record (e.g. anomalous dry conditions), the environment-to-circulation approach will be used to investigate potential climate mechanisms associated with drought conditions reconstructed from sedimentary proxy data from Long Lake, Wyoming. This paper focuses on the relationship between atmospheric synoptic processes and surface climate mechanisms and their influence on the ecological changes recorded during the *Populus* period identified by Carter et al., (2013; 2017a).

## 1. Study Area

Long Lake (41° 30.099′ N, 106° 22.087′ W; 2700 m a.s.l.) is located within the Upper Platte River watershed in the Medicine Bow Range of south-eastern Wyoming (Figure 1). The lake lies within a closed drainage basin behind a Pinedale-aged terminal moraine with no inlets or outlets (Atwood, 1937). The study site experiences a snow-dominated winter precipitation with a precipitation maximum in May (Mock, 1996; Shinker, 2010), albeit that May precipitation maximum only accounts for 12-15% of the total annual precipitation (Shinker, 2010). Interpolated modern January and July precipitation and temperatures from the nearest weather station suggests an average of 330 and 690 mm, and -9.7°C and 11°C, respectively (NRCS, unpublished data). Using the modern pollen analogue technique (Overpeck et al., 1985), Carter et al. (2017a) reconstructed the mean temperature of the coldest month (MTCO; i.e. January) , mean temperature of the warmest month (MTWA; i.e. July) , and annual precipitation which have averaged -9°C, 15°C, and ~443 ± 39 mm over the past ~2000 years. During the drought between 4300 and 4100 cal yr BP, MTCO, MTWA and annual precipitation averaged -8°C, 16°C, and 394 ± 58 mm (Carter et al., 2017a) Both the modern and reconstructed climate from the area highlights that the Medicine Bow Range has a high degree of precipitation variability, likely related to natural fluctuations in the strength and position of the jet stream. Currently, the study region does not experience seasonal precipitation patterns associated with ENSO phases (Wise, 2010; Heyer et al., 2017).

Modern vegetation at the study site is comprised mostly of lodgepole pine (*Pinus contorta*), with Engelmann spruce (*Picea englemanni*) and subalpine fir (*Abies lasiocarpa*) on more mesic soils. Currently, the modern geographical location of the aspen ecotone is roughly 200 m a.s.l. downslope from Long Lake (Carter et al. 2017a). Lodgepole pine has been the dominant canopy cover type for the past ~8000 years (Carter et al., 2013). However, the region experienced a rapid change in vegetation composition from a lodgepole pine dominated forest to a mixed forest of lodgepole pine and quaking aspen between ~4000 and 3450 cal yr BP in response to drought conditions centred on 4200 cal yr BP (Carter et al., 2017).

Carter et al. (2013; 2017a) describe the sedimentary collection that took place at Long Lake in 2007, while Carter et al. (2017a) describe age-depth relations, charcoal and pollen analysis, and the modern pollen analogue technique used to reconstruct local temperature and precipitation values. Additionally, Carter et al. (2017a) also updated the age-depth relations from Carter et al.

(2013) with the additions of an AMS radiocarbon date that was used to temporally constrain the upper and lower ages of the '*Populus*' period, as well as temporally constraint the drought at 4200 cal yr BP (see Table 1, Carter et al., 2017a).

## 2. Methods

### Modern climate analogues and calculation of composite-anomaly values

In order to identify potential climate mechanisms associated with the changes in vegetation composition at Long Lake, Wyoming, a time series of modern precipitation anomalies was calculated from Wyoming Climate Division 10, the Upper Platte River basin using data from the National Climate Data Centre (NCDC) (Figure 2). The Upper Platte River basin was chosen because it encompasses the Medicine Bow Range where Long Lake and sedimentological analyses were collected from (Carter et al., 2013; 2017a). Annual average precipitation values from 1979-2014 were compared to the long-term mean (1981-

2010) to create a time series of annual precipitation anomalies. The time series was calculated for 1979-2014 which is the common period for the North American Regional Reanalysis (NARR) dataset. From the time series, potential analogues (e.g. dry case years) that were identified to be the most statistically dry years were selected to represent similar conditions (e.g. anomalously dry) that would have potentially caused the ecological response identified in the sedimentary analyses conducted by Carter et al. (2013; 2017a) . In this study, case years that were -1 standard deviations below the long-term average (one

standard deviation being equivalent to 58.89 mm per year) were selected as modern analogues Statistical significant of each case year were evaluated at each grid point in our study region using a two-tailed Student's t-test which an alpha of 0.05 to quantify significance. Precedence for using a t-test to calculate statistical significance of anomalies in climatological analyses is well established in the existing literature (Cayan, 1996; Shabbar and Khandekar, 1996; Taschetto and England, 2009). The results are presented in a map depicting the spatial distribution of significant p-values across our study region of south-western

Wyoming (Figure 2b).

   For the purpose of understanding modern synoptic processes that may have influenced vegetation change, a variety of surface and atmospheric climate variables from the NARR dataset (Mesinger et al., 2006) were used to calculate and map anomalous patterns (Table 1). The NARR dataset is advantageous for two reasons; 1) it provides a variety of climate variables that

represent atmospheric synoptic processes (e.g. atmospheric pressure, wind direction and speed, moisture availability, and vertical motion), as well as surface conditions (e.g. precipitation rate and temperature); and 2) The spatial resolution (32-km grids) of the NARR dataset is at a finer scale than large-scale GCMs making the NARR dataset useful for assessing hydroclimate impacts at high spatial resolution (Heyer et al., 2017). The 32-km resolution is therefore valuable for capturing the topographic and climate diversity of the geographic study region.. The seasonal values (e.g. winter = December, January,

and February; or DJF) of the selected modern analogue case years were averaged together (composited) and compared to the long-term mean (1981-2010) to create composite-anomaly values for each season. These composite-anomaly values were mapped in order to analyze and assess the spatial and temporal variability of both the surface and atmospheric variables to

identify conditions that would support vegetation change identified in the sedimentary record. Surface variables were mapped at a regional level to illustrate the spatial heterogeneity of such processes. Atmospheric variables were mapped at a continental scale to illustrate the large spatial scales in which such variables operate. Composite-anomaly values were calculated using the NAAR Monthly/Seasonal Climate Composites plotting and analysis page (https://www.esrl.noaa.gov/psd/cgi-bin/data/narr/plotmonth.pl). The resultant netCDF (Network Common Data Form) files were plotted graphically using the NASA/GISS software, Panoply, a netCDF data viewer (https://www.giss.nasa.gov/tools/panoply/). The resulting maps are plotted using the NARR 32-km gridded format and have not been interpolated in order to maintain the native spatial representation of the data.

### 3. Results

Five case years (2012, 2002, 2001, 1988, and 1994) that were -1 standard deviations below the long-term average were chosen because they were found to be the most statically significant ($p < 0.05$) analogues for dry conditions in the Upper Platte River basin. These five case years are representative of persistently low annual precipitation conditions and are therefore used as analogues to explain the climate mechanisms responsible for dry conditions identified in the paleoclimate record (Carter et al., 2013; 2017a). The composite-anomaly values provide context regarding the spatial patterns associated with the anomalously dry conditions.

**4.1 Modern climate analogues of extreme dry conditions in the Upper Platte River Basin**

**4.1.1 Surface modern climate analogues**

The composite-anomaly maps for precipitation rate provide the spatial representation of the information shown in the time series of annual precipitation. Winter (DJF) composite-anomaly values for precipitation are slightly above normal in the study region (Figure 3a). However, based on the time-series, the overall annual average was lower-than-average throughout the case years. Spring (MAM) composite-anomaly values for precipitation rate indicate a shift toward slightly drier-than-normal conditions (Figure 3b). Summer (JJA) composite-anomaly values indicate an increase in aridity (Figure 3c) that persisted in the region through the fall (SON) (Figure 3d).

Seasonal composite-anomaly maps for temperature (Figure 4) provide information on local surface conditions during the anomalous case years. Winter (DJF) composite-anomaly values are slightly cooler-than-normal in the region (Figure 4a). Positive temperature anomalies increased during the spring, and persisted from summer into the fall (Figure 4d).

**4.1.2 Atmospheric modern climate analogues**

In the atmosphere, 500mb geopotential height anomalies are aligned with surface temperature anomalies. Figure 5 illustrates 500mb geopotential height composite anomalies during the anomalous case years, which is shown on a continental scale because it captures regions of lower-than-normal atmospheric pressure (associated with enhanced troughs) and higher-than-normal atmospheric pressure (associated with enhanced ridges). Winter (DJF) composite-anomaly values for 500mb geopotential height show slightly lower-than-normal pressure centred over the interior of Canada and extending down over the Great Plains, and higher-than-normal pressure in the north Pacific (Figure 5a). Spring (MAM) composite-anomaly values indicate a higher-than-normal pressure centred over the central Great Plains (Figure 5b), which shifted north over the northern Great Plains during the summer (JJA) (Figure 5c). Fall (SON) composite-anomaly values show slightly higher-than-normal pressure over the western US (Figure 5d).

The 500mb vector wind composite anomaly maps provide information on the anomalous component of flow (Figure 6), which is associated with the 500mb geopotential height composite-anomaly maps. For winter (DJF), the anomalous component of flow is northerly into the study region (Figure 6a). During the spring (MAM), the anomalous component of flow is from the south- to south-east (Figure 6b) associated with the clockwise flow of air around the anomalous ridge seen in Figure 5b. During the summer (JJA), the anomalous component of flow is from the east (Figure 6c) associated with the clockwise flow of air around the anomalous ridge seen in Figure 5c. For fall (SON), the anomalous component of flow is northerly into the study region (Figure 6d).

While the 500mb geopotential height and 500mb vector wind composite-anomaly maps provide a continental perspective of broad-scale anomalous ridges and troughs and subsequent advection of wind, 500mb Omega (vertical velocity) offers a more local-scale perspective on secondary sinking and rising motions that occur within ridges and troughs, respectively. Specifically, positive 500mb Omega composite-anomaly values show anomalous sinking motions, indicating suppression of precipitation, and negative 500mb Omega composite-anomaly values indicating anomalous rising motions that enhance precipitation (Figure 7). Winter (DJF) composite-anomaly maps for 500mb Omega indicate weak anomalous rising motions in the study region (Figure 7a). Spring (MAM) and Fall (SON) composite-anomaly maps indicate strong anomalous sinking motions in the study region (Figure 7b and 7d). Summer (JJA) composite-anomaly maps show a mixture of anomalous weak rising and sinking motions in the study region (Figure 7c).

The 850mb specific humidity composite anomaly values were plotted on a continental scale to provide context on the spatial extent of atmospheric moisture available for uplift by Omega during each season (Figure 8). Winter (DJF) 850mb specific humidity composite anomaly-values are slightly below normal in the study region (Figure 8a). Spring (MAM) 850mb specific humidity composite-anomaly values indicate below normal moisture availability (Figure 8b), which persisted into the summer (JJA) and fall (SON) (Figure 8c, d).

## 5. Discussion

### 5.1 Proxy evidence for regional climate variability at 4200 cal yr BP

Ecological responses can vary spatially across the central Rocky Mountains and central Great Plains because of different synoptic controls. For example, Carter et al. (2013; 2017) found palynological evidence that documents a shift in vegetation composition beginning around 4000 cal yr BP at Long Lake, Wyoming which was attributed to a ~150-year long drought between 4300 and 4100 cal yr BP. While the authors were unable to confirm whether the drought was indeed 150-years long, multi-decadal- to centennial-scale droughts were common phenomena in the Great Plains and western US during the late Holocene (Woodhouse and Overpeck, 1998; Cook et al., 2004; Schmieder et al., 2011; Cook et al., 2016). Regardless, the timing of the drought reconstructed from Long Lake, Wyoming occurred during similar conditions identified over the Great Plains region (Booth et al., 2005). Several sites near Long Lake also experienced ecological changes associated with drought conditions; first, clusters of optical luminescence and radiocarbon dates ~4200 cal yr BP suggest widespread dune reactivation in dune fields in Wyoming and in the central Great Plains (Stokes and Gaylord, 1993; Mayer and Mahan, 2004; Halfen et al., 2010). Similarly, Miao et al. (2007) also demonstrate active aeolian activity between 4500 and 2300 cal yr BP, however, the authors acknowledge they cannot rule out aeolian responses lagged to climate (Miao et al., 2007); second, high concentrations of sand influx between 4200 and 3800 cal yr BP also indicates drought conditions in the Sand Hills of Nebraska (Schmieder, 2009); third, lower lake levels were recorded between ~5000 and ~3400 cal yr BP from several lakes in the central Rocky Mountains near Long Lake, further supporting regionally dry conditions during this time period (Shuman et al., 2014; Shuman et al., 2015); lastly, reconstructed paleoclimate data from the mid-latitudes of North America (i.e. northern Great Plains) suggest extensive drought conditions persisted between 4700 and 4000 cal yr BP (Shuman and Marsicek, 2016). However, this was a not predominant feature in their record. Widespread and severe drought conditions were also recorded across the Rocky Mountains and Great Plains region between 4700 and 4000 cal yr BP based on pollen, charcoal, diatom, grain-size analysis, testate amoebae assemblages, and speleothem stable isotopes, (Dean, 1997; Bernal et al., 2011; Schmieder et al., 2011; Lundeen et al., 2013; Morris et al., 2013; Wanner et al., 2015; Carter, 2016). Yet, the exact timing of drought conditions based on different proxy varies spatially and temporally.

Conversely, cool and wet conditions have also been proposed throughout North America between 5500 – 3800 cal yr BP based on stable isotopes, pollen, tree-ring data, as well as by advances in glaciers (Menounos et al., 2008; Grimm et al., 2011; Mayrer et al., 2012; Anderson et al., 2016). Namely, Grimm et al. (2011) did not record extensive drought conditions at 4200 cal yr BP in the northern Great Plains. Rather, the authors suggest a regime shift towards wet conditions around 4400 cal yr BP, which counters the finding presented by Booth et al. (2005). Similarly, in south-central Colorado, negative excursions in $\delta^{18}O$ values from Bison Lake, Colorado ~4200 cal yr BP, coupled with increases in spruce (*Picea*) pollen are interpreted as being

indicative of colder-than-previous temperatures and increased snowfall (Anderson, 2012; Anderson et al., 2015; Anderson et al., 2016). Lastly, treeline abruptly declined ~4200 cal yr BP in the Great Basin region further suggesting cool conditions (Salzer et al., 2014). Such variability in climate reconstructed from proxy data across North America could be a function of site or proxy sensitivity, response time, and temporal resolution making it difficult to pinpoint the exact timing and spatial extent of major climatic shifts experienced during this time period (Schmieder, 2009). For example, Higuera et al. (2014) suggest that significantly shorter fire return intervals between 4000 and 3500 cal yr BP in Rocky Mountain National Park correspond with above-average lake levels from Hidden Lake, Colorado (Shuman et al., 2009). The authors suggest that higher lake levels indicate snow-dominated winters, but increased fire activity indicates drier summers. However, variability in climate may also, represent surface climate responses to both large-scale changes in the polar jet stream, as well as small-scale controls such as topography (Barry, 1970; 1982; Mock, 1996; Shinker, 2010; Mock and Shinker, 2011) that are inherent to the heterogeneous climate throughout the topographical complex interior intermountain west.

The spatial and temporal heterogeneity in proxy data during this time period is likely the result of broad-scale reorganization in climate. Specifically, the time period between 5000 and 4000 cal yr BP is when the onset and intensification of the El Niño Southern Oscillation (ENSO) occurred (Shulmeister and Lees, 1995; Barron and Anderson, 2010), as well as a switch from a more negative Pacific North-American (PNA) phase (i.e. more enhanced zonal circulation) to a more positive PNA phase (i.e. more enhanced meridional circulation) between 4200 and 4000 cal yr BP (Fisher et al., 2008; Anderson et al., 2016; Liu et al., 2014). Both ENSO and the PNA are primary controls of modern winter climate variability in some parts of North America (Müller & Roeckner, 2006; Notaro et al., 2006; Allen et al., 2014), although impacts of ENSO in our study region of south-eastern Wyoming are minimal (see Heyer et al., 2017 and Wise, 2010). Positive PNA patterns are typically associated with more meridional flow, positive winter temperature and negative precipitation anomalies over the Pacific Northwest, and have been linked to wildfire activity in the Southern Canadian Rocky Mountains (Wallace & Gutzler, 1981; Leathers et al., 1991; Fauria & Johnson, 2008; Allen et al., 2014). Together, these two modes of variability can influence the position of the jet stream which subsequently influences both modern, and likely past regional temperature and precipitation in certain parts of western North America.

Another source of seasonal precipitation variability in the west is associated with the North American Monsoon system (NAM), albeit largely in the southwest portion of North America (Adams and Comrie, 1997, Mock 1997, Shinker 2010). A weakening of the NAM system is proposed between 5000 and 4000 cal yr BP (Metcalfe et al., 2015). While our study region occasionally benefits from advection of moisture recycled from the southwest (Dominguez et al., 2009), the overall atmospheric circulation controls within the central Rocky Mountains is still dominated by westerly winds via the polar jet stream even in summer (Mock 1996; Shinker 2010) versus the shift in circulation-driven winds in the southwest associated with the NAM (Adams and Comrie, 1997).

Imbedded within the period of climate organization described above was the '4.2 ka event' which was a prominent dry period found primarily at low-to-mid latitudes and was responsible for cultural collapses globally (deMenocal, 2001; An et al., 2005; Weiss, 2016; 2017a; 2017b). This climatic event has been suggested to be the formal boundary between the mid- and late-Holocene (Walker et al., 2012). Currently, there is no clear mechanistic explanation behind the 4.2 ka event (Walker et al., 2012), but several hypotheses exist. The first hypothesis is that the 4.2 ka event was the result of Bond event 3 (Bond et al., 1997; 2001). Yet, there is currently no precise mechanistic explanation for the Bond cycles (Wanner et al., 2014). The second hypothesis is that the drought was caused by the general southward migration of the Intertropical Convergence Zone (ITCZ) due to decreased late-Holocene summer/annual insolation (Liu et al., 2014). A southward migration of the ITCZ offers a potential climatic mechanism because of its influence on the position of the jet stream, which as previously discussed, significantly impacts North American winter temperature and precipitation patterns (see Mock, 1996). Finally, persistent La Niña-like conditions have been proposed as one of the hypothesized causal factors of drought centred on 4200 cal yr BP. La Niña-like conditions have been linked to other severe and prolonged droughts during the Holocene (Booth et al., 2005; Forman et al., 2001; Menking and Anderson, 2003), as well as the Dust bowl drought in the 1930s (Schubert et al., 2004), and the recent drought between 1998 and 2002 (Hoerling and Kumar, 2004). However, the prescription of persistent La Niña-like conditions does not address the atmospheric processes at a local and regional scale that may have led to the widespread mega drought conditions centred on 4200 cal yr BP. By identifying extreme dry years from modern precipitation data near Long Lake, the modern climate analogue technique is used here to identify atmospheric circulation mechanisms that supported hydrologic extremes in the modern record as analogues for synoptic climate processes of past hydrologic extremes evident in the pollen record via the environment-to-circulation approach (Mock and Brunelle-Daines, 1999; Shinker et al., 2006; Shinker, 2014).

## 5.2 Regional climate variability based on modern climate analogues of drought at 4200 cal yr BP

Modern climate analogues in this paper illustrate slightly cooler-than-normal and slightly wetter-than-normal winter conditions in the Medicine Bow Range (Figures 3 and 4). Cooler-than-normal winter temperatures during the winter months can be explained by several factors; first, cold and dry air from the interior region of Canada was drawn to the study region by anomalous flow associated with the anomalous high-pressure ridge centred off the coast of the Pacific Northwest (Figures 5a, 6a); and second, the slightly higher-than-normal precipitation (Figure 3a) likely increased the possibility of greater-than-normal cloud cover. Slightly wetter-than-normal during the cool season in mountainous regions are likely a result of local orographic uplift (Mock, 1996; Shinker, 2010), demonstrated by anomalous rising vertical motions in the study area (Figure 7a). However, while winters were slightly wetter-than-normal, While the overall annual precipitation values for the selected case years were lower-than-average in all seasons of all case years, with the exception of a couple of seasons used to calculate the composite-anomaly values. Overall, the interior intermountain west experiences both within-year and between-year

variability of precipitation (Mock, 1996; Shinker, 2010). This within- and between-year variability is likely a result of variations in the polar jet during winter months.

Winter conditions at Long Lake are currently not impacted by phases of ENSO (Heyer et al., 2017, Wise 2010), as it is positioned within the transition zone (between 40° – 42°N) that include consistently low correlation values between Pacific sea-surface temperature anomalies and cool season precipitation (Dettinger et al., 1998; Wise, 2010; Heyer et al., 2017). Analysis of this transition zone of low correlation values between sea-surface temperature anomalies and cool season precipitation over the past 500 years suggest it has been stable ~40°N latitude (Wise, 2016). Paleoecological studies from the study region have also suggested a relatively stable transition zone throughout the Holocene (Carter et al., 2013; Mensing et al., 2013). Thus, while Barron and Anderson (2010) concluded an enhanced ENSO pattern c. 4.0 ka BP may have been associated with an increase in winter precipitation in the southern Rocky Mountains (Anderson et al., 2012), it is likely that the enhanced ENSO pattern contributed to an increase in variability of the polar jet stream (Heyer et al., 2017) creating spatial inconsistencies of winter precipitation anomalies in the region in the past. Based on the modern climate analogues presented here, winter conditions are supportive of proxy data demonstrating the time period between 5000 and 4000 cal yr BP as being climatically variable likely in response to variations in the polar jet stream. Furthermore, our selected modern analogue case years represent a mixture of ENSO modes (e.g. El Niño, La Niña and Neutral conditions) indicating that these modes of variability, while statistically significant in the Southwest (as described by Wise, 2010 and Heyer et al., 2017) do not impact our study area in a consistent manner.

While winter precipitation is beneficial for vegetation during the growing season in the form of soil recharge via snowpack accumulation, peak precipitation maximum in the study region of south-eastern Wyoming occurs during the late spring (i.e. May) (Mock, 1996). Thus, changes in late spring/early summer conditions are more likely to impact vegetation and soil recharge in the study area. This is demonstrated by modern climate analogues which illustrate anomalously warm and dry conditions beginning in the spring and persisting throughout the growing season (Figures 3 and 4) not only at Long Lake, but also across the entire west-central Great Plains. The warmer-than-average temperatures are directly related to an anomalous and persistent high-pressure ridge centred over the central Great Plains region in the spring, which persists over the northern Great Plains and study region during the summer (Figures 5b, c). As a result of anomalous anti-cyclonic winds, dry continental air from the interior of North America were delivered into the study region (Figure 6b, c). The delivery of anomalously dry air (Figure 8b) into the study region in conjunction with enhanced local sinking motions in the atmospheric (Figure 7b) ultimately suppressed precipitation in the spring. While some anomalous rising motions were present in the summer via 500mb Omega values (Figure 7c), there was lower-than-normal moisture in the atmosphere (via 850 mb specific humidity) to be uplifted and precipitated (Figure 8c). The lack of atmospheric moisture further supported the enhancement of drought conditions in the study region during the growing season.

Typically, southerly winds known as the Great Plains low-level jet (Schmeisser et al., 2010) are responsible for bringing in moist air from the Gulf of Mexico to the Great Plains region during late spring/early summer (Wilhite and Hubbard, 1998; Sridhar et al., 2006). These southerly winds are associated with the anti-cyclonic flow around the Bermuda High off the coast of eastern North America. However, if the Great Plains low-level jet is closed off from its moisture source, the Gulf of Mexico,

the Great Plains region will essentially be dry during the summer months. Schmeisser et al. (2010) suggest that in order for drought to develop and persist in the Great Plains region during the summer months, the Bermuda High must be reduced or positioned either more easterly or southerly, which would create a more south-westerly component of flow. Change and Smith (2000) suggest several other factors are involved with drought in the Great Plains region. First, the prominent feature is an anticyclone positioned over the central portion of North America; second, the midtropospheric westerly winds weaken and

become easterly winds in association with the anti-cyclonic high-pressure positioned over the Great Plains; and third, the Bermuda high-pressure has a westward displacement rather than a reduced or more easterly or southerly position, as suggested by Schmeisser et al. (2010). This westward displacement of the Bermuda high-pressure causes the enhancement of a low-level warm flow into the central Great Plains region causing the region to experience negative relative humidity anomalies. The 500mb geopotential height composite-anomaly maps (Figure 5b) for both spring and summer illustrate a high-pressure ridge

centred over the central Great Plains in the spring, and shifting north during the summer (Figure 5c). As a result, midtropospheric winds, as seen in 500mb vector winds (Figure 6b, c), illustrate an easterly- to south-easterly component of flow around the anomalous high-pressure ridges which likely inhibited growing season moisture from the Gulf of Mexico via the low level jet, which is especially important for dune stabilizing grasses and vegetation in the central Great Plains (Schmieder et al., 2011). Modern climate analogues visibly illustrate a reduced, yet northward displacement of the western

ridge of the Bermuda high-pressure which likely contributed to dry conditions in the region. And finally, modern climate analogues clearly illustrate a lack of relative humidity in the western Great Plains and study region of south-eastern Wyoming (Figure 8b, c). These results offer a climatic explanation that resulted in the ecological changes ~4000 cal yr BP, as recorded in the sedimentary data at Long Lake, as well as provide a mechanistic explanation regarding the reactivation of dunes in Wyoming and the central Great Plains during this time.

**5.3 Model simulations and implications for drought in the central Rocky Mountains and central Great Plains**

The current state of research is in agreement that anomalous and persistent high pressure ridges over the Great Plains are the most common contributor of drought (Basara et al., 2013). Persistent high pressure ridges lead to subsidence (e.g. sinking

vertical motions) which suppresses precipitation. Persistent high pressure ridges also prevent the typical southward movement of cold fronts from Canada which serve to organize spring rains, block delivery of moisture from the Gulf of Mexico, as well as inhibit convective thunderstorms which contribute to summer precipitation in the Great Plains region (Hoerling et al., 2014). Using a complex numerical weather-predication model with data from May 1987 to May 1988, Palmer and Brankovic (1989) had significant skill in predicating an anomalous high pressure ridge over North America during the summer of 1988. However,

there is less agreement on the boundary conditions required to initiate anomalous and persistent high pressure ridges over the Great Plains. In particular, the relationship between Pacific and Atlantic teleconnections and Great Plains drought is not very well understood (Basara et al., 2013). While it has been suggested from both modern observation and modelled data that a relationship exists between SSTs and drought (Trenberth et al., 1988; Palmer and Brankovic, 1989; Kalnay et al., 1990; Schubert et al., 2004; Feng et al, 2008; Basara et al., 2013), Hoerling et al. (2014) found weak evidence to support SST as a strong forcing on major droughts in the central Great Plains because droughts occurred in each phase of ENSO.

While our study is not able to address, nor model boundary conditions involved in initiating drought within the study region of south-eastern Wyoming ~4200 cal yr BP, the process-based approach of the modern climate analogue can be used to inform future paleoclimatic modelling and drought prediction. For example, as discussed above, both modern observations and simulations have demonstrated that anomalous and persistent high pressure ridges over the Great Plains are important synoptic processes involved in drought conditions over the Great Plains region. Similarly, enhanced anti-cyclonic circulation over the Great Plains was found to be a prominent feature causing mid-Holocene droughts in the region (Diffenbaugh et al., 2006). Thus, using the underlying assumption involved in the modern climate analogue, the principle of uniformitarianism (Barry and Perry, 1973), our results suggest that high pressure ridges and anti-cyclonic circulation over the Great Plains region, likely contributed to the drought identified at 4200 cal yr BP. Based on the geographical proximity of our study region to the central Great Plains region, we hypothesize that severe and persistent droughts have the ability to affect the eastern most parts of the central Rocky Mountains.

## 6.  Conclusion

Paleoecological reconstructions are valuable for understanding how ecosystems and disturbances respond to both gradual and abrupt changes in climate. However, proxies preserved in sedimentary records fail to record mechanisms that caused the ecological responses; the proxies only record that there was a change in vegetation composition or a change in disturbance regimes. Using the modern climate analogue technique and the underlying principle of uniformitarianism, our results offer potential climatic mechanisms that explain how persistent drought conditions in the central Great Plains affected vegetation composition at Long Lake, Wyoming ~4000 cal yr BP. Specifically, the atmospheric modern climate analogues illustrate persistent anomalous high pressure positioned over the Great Plains region, which coincidently have been associated with the most recent droughts of the 20[th] century, regardless of ENSO mode or any other mode of variability. While the modern case years show weak rising motions during the summer months, there wasn't enough moisture available in the atmosphere via specific humidity at 850mb for precipitation anomalies to occur. In other words, the mechanism for uplift was present over the study region, but the moisture availability was not present as a result of a persistent high-pressure ridge which drew warm and dry air into the study region in a southerly- to south-easterly direction from the interior region of North America. Additionally, the persistent high-pressure ridge positioned over the Great Plains during the growing season likely interrupted the normal

moisture flow from the Gulf of Mexico via the low-level jet into the region, instead drawing warm and dry air from the interior of North America to the study region. The combination of higher-than-normal geopotential heights, anomalous component of flow and lack of moisture transport to the study region created the anomalously warm and dry conditions in the study region, thus providing a potential mechanistic explanation for anomalous dry conditions that support the ecological change,
widespread dune reactivation, and lowered lake levels associated with the drought at 4200 cal yr BP in the central Rocky Mountains and central Great Plains (Halfen et al., 2009; Mayer and Mahan, 2004; Stokes and Gaylord, 1993; Shuman et al., 2015).

Many reconstructions using proxy data from sites within the intermountain west (especially in the western Rocky Mountains) 
and northern Great Plains indicate a variety of conditions around 4200 cal yr BP (Menounos et al., 2008; Grimm et al., 2011; Mayrer et al., 2012; Anderson, 2012; Anderson et al., 2016). However, because Long Lake, Wyoming is the eastern most record in the central Rocky Mountains with the closest geographic proximity to the central Great Plains, results from Long Lake indicate synchronicity of drought signal during 4200 cal yr BP with the central Great Plains. The anomalous high pressure centred over the central Great Plains likely has little influence in terms of both the northern Great Plains where influences in 
ground water are important (Grimm et al., 2011), and farther west within the Rocky Mountains where a high degree of spatial and temporal variability of seasonal precipitation occurs driven by fluctuations in the polar jet stream (Mock, 1996; Shinker, 2010). For example, Shinker (2010) illustrated the heterogeneity of precipitation in the interior intermountain west by assessing the monthly contribution of annual precipitation within the region. Even with a lack of distinct precipitation seasonality in any given month within the interior intermountain west, high elevation precipitation events (or lack of) can easily offset (or 
enhance) water deficit (Shinker 2010). Such variability of precipitation within the interior intermountain west, driven by variations in the strength and position of the jet stream, help to explain the inconsistency in drought across the region. Finally, Long Lake experiences similar seasonal precipitation characteristics as the central Great Plains (see Shinker, 2010) rather than sharing seasonal characteristics of other interior intermountain west sites to the west, the northern Great Plains, and the desert southwest. While an out of phase relationship with the desert southwest and Great Plains region has been proposed (i.e. wetter 
monsoons during drier Great Plains) (Dominguez et al., 2009), our results do not illustrate enhanced monsoon activity. This would agree with previous literature (Metcalfe et al., 2015) suggesting the weakening influence of NAM outside of the true monsoon region (i.e. Arizona, New Mexico, and north-western Mexico region) ~4000 cal yr BP.

   Droughts such as the one centred on 4200 cal yr BP, as well several droughts in the 13[th] and 16[th] centuries were more severe 
and of longer duration than the more recent droughts of the 20[th] century (Woodhouse and Overpeck, 1998).  Understanding the climate processes associated with modern drought provide better insight of past drought variability and the mechanisms that caused mega droughts evident in paleoecological records. This study demonstrated the benefits of applying a modern climate analogue technique to the paleoecological reconstruction from Long Lake in order to better understand the potential

climatic mechanisms that impacted ecological changes such as the changes in quaking aspen populations in the Medicine Bow Range.

## 7. Data Availability

Modern climate analogue years were selected based on NOAA/NCDC Wyoming Climate Division 10 from the Earth System Research Laboratory Physical Science Division of NOAA (https://www.esrl.noaa.gov/psd/data/timeseries/). Surface and atmospheric variables used in the composite-anomaly analysis are available through the Earth System Research Laboratory Physical Science Division of NOAA (https://www.ncdc.noaa.gov/data-access/model-data/model-datasets/north-american-regional-reanalysis-narr). Pollen and charcoal data have been uploaded to the Neotoma Paleoecology Database web page: https://www.neotomadb.org/groups/category/pollen; https://apps.neotomadb.org/Explorer/?datasetid=24878. Pollen and charcoal data are interpreted at 1-cm resolution between depths 94 and 176 cm, as described by Carter et al. (2017).

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

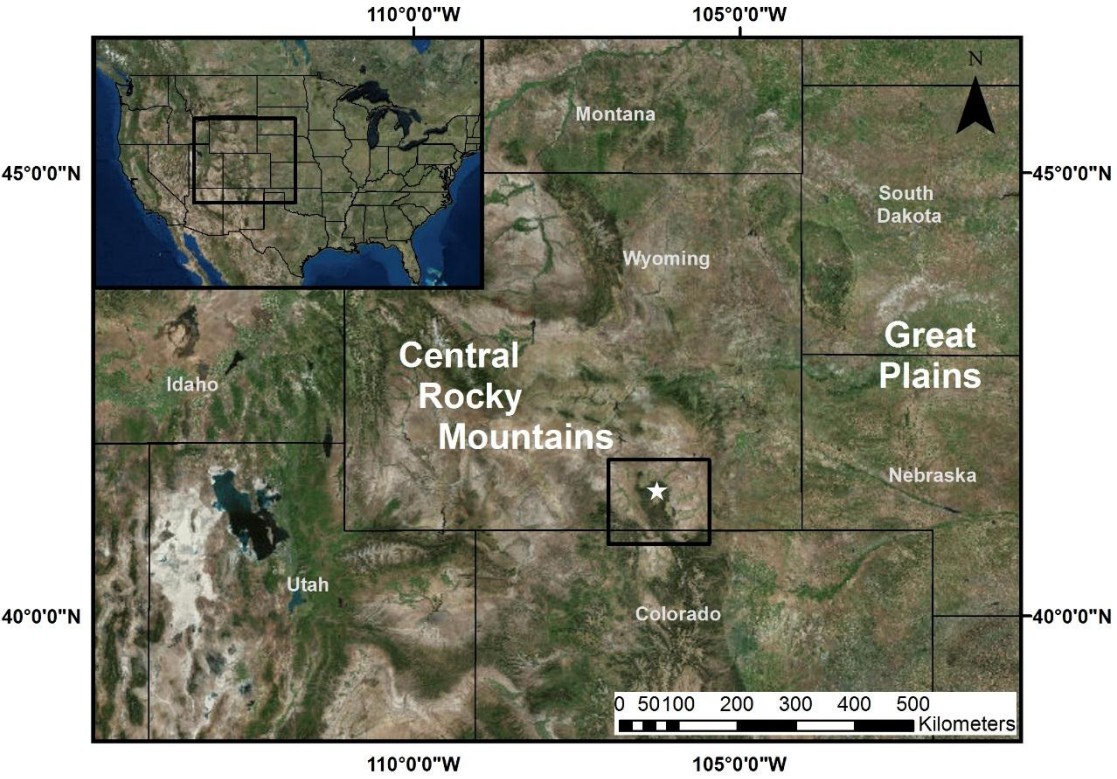

**Figure 1. Location map of the study region in the western United States (small panel; black box). Long Lake, Wyoming (white start inside the black box indicating the study area) is located in south-eastern Wyoming within the central Rocky Mountain region on the edge of the Great Plains.**

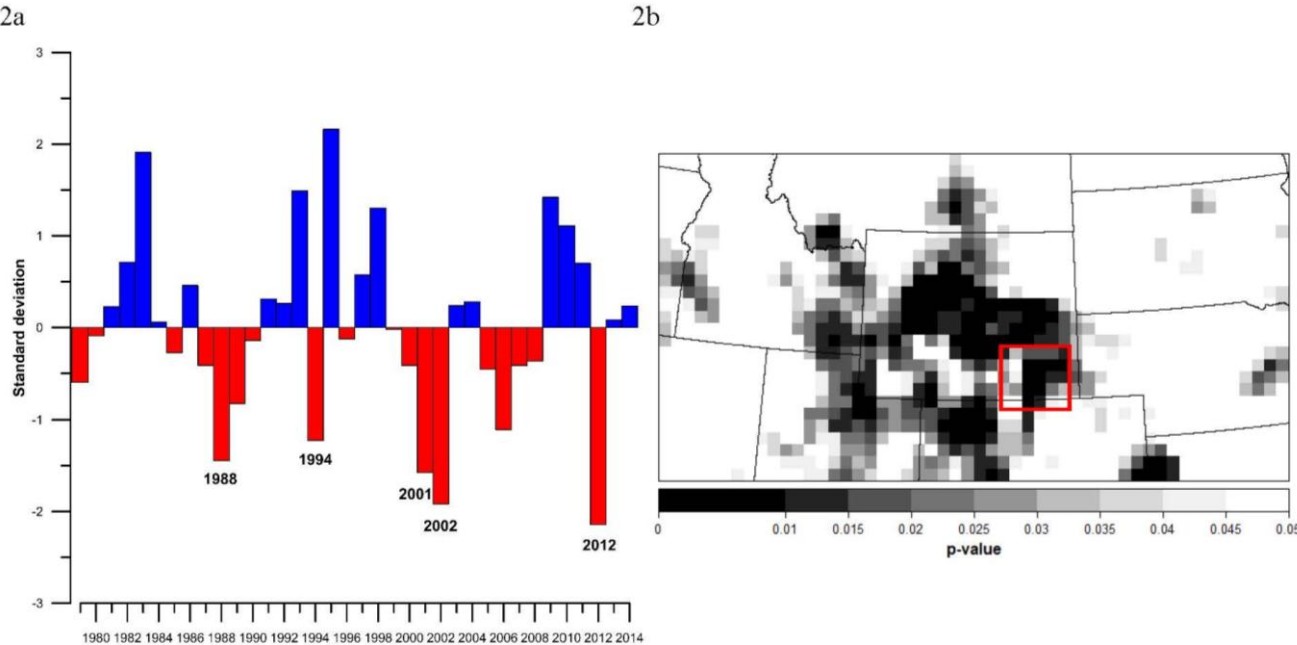

2a 2b

**Figure 2. Precipitation anomalies and the spatial distribution of significant p-values across the study region of south-eastern Wyoming. A) Time series of annual precipitation anomalies for 1979-2014 compared to the long-term average (1981-2010) from Wyoming climate division 10, Upper Platte River Basin. The first five years with -1 or more standard deviations below the long-term average include 2012, 2002, 2001, 1998, and 1994. One standard deviation equates to 58.89 mm. Climate division data were collected from http://www.esrl.noaa.gov/psd/cgi-bin/timeseries/timeseries1.pl. B) Map showing the spatial distribution of significant p-values ($p < 0.05$) across the study region (outlined in red box) identified during the five driest years. P-values were evaluated using a two-tailed Student's t-test with an alpha of 0.05.**

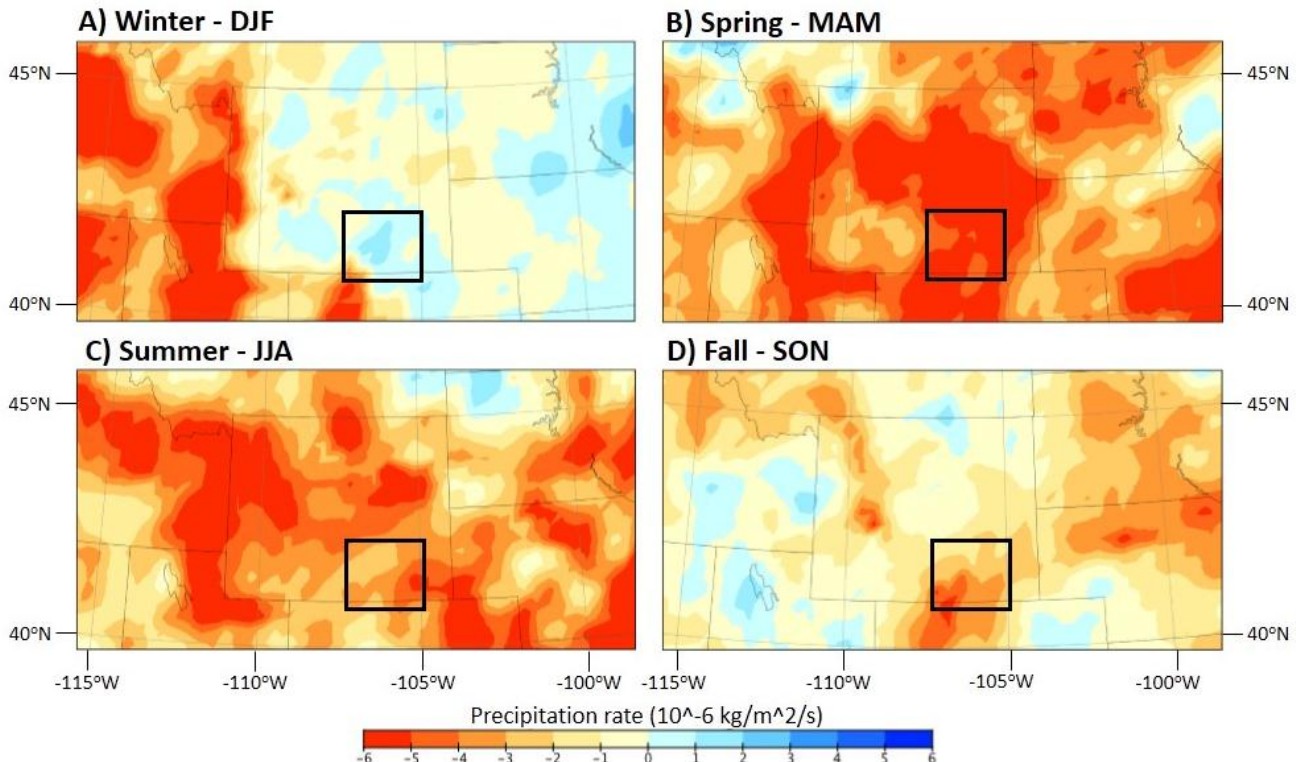

**Figure 3. Composite anomaly maps for precipitation rate at the surface. A) Precipitation rate at the surface for winter (DJF); B) Spring (MAM); C) Summer (JJA); D) Fall (SON). Positive values (cool colours) for precipitation rate indicate wetter-than-normal conditions. Negative values (warm colours) indicate dryer-than-normal conditions. The black box denotes the study site, Long Lake in the Medicine Bow Mountains of south-eastern, Wyoming. Light grey lines depict lines of latitude/longitude.**

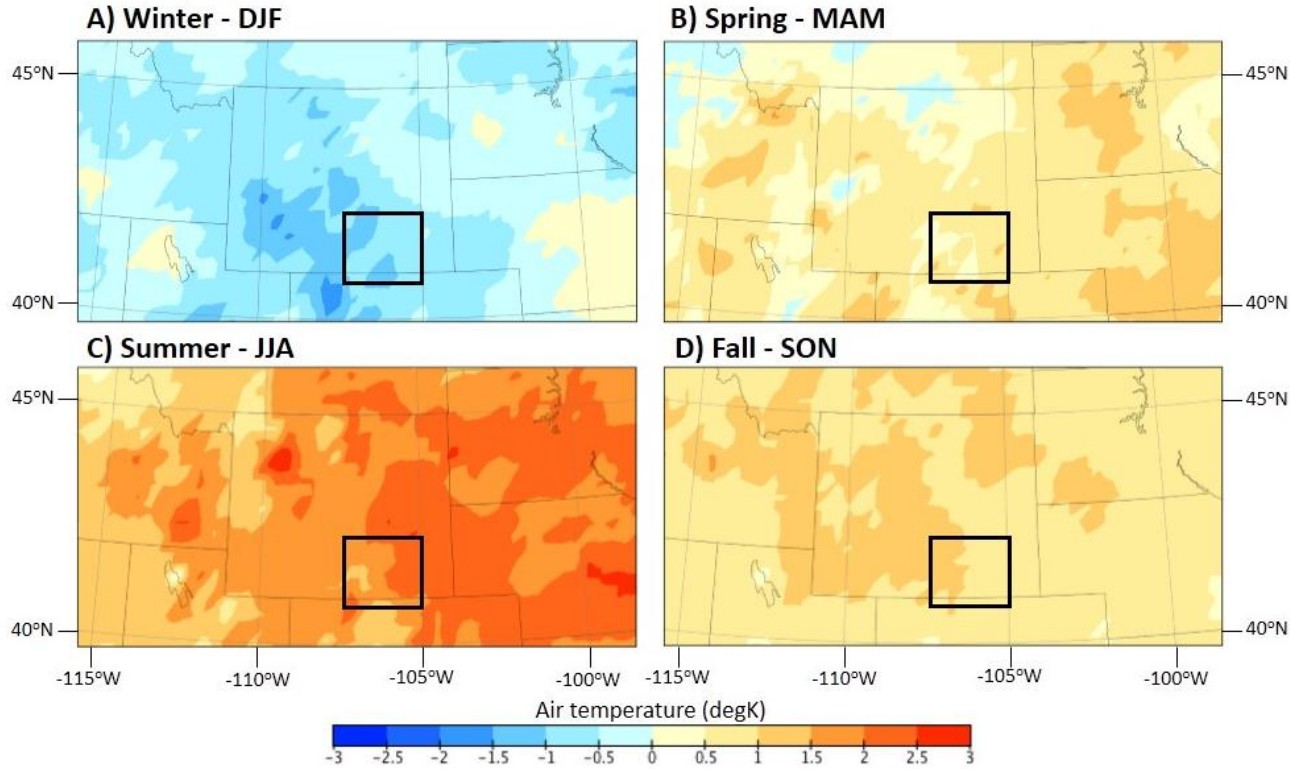

**Figure 4. Composite anomaly maps for air temperature at the surface. A) Air temperature during the winter (DFJ); B) Spring (MAM); C) Summer (JJA); D) Fall (SON). Positive values (warm colours) for air temperature indicate warmer-than-normal conditions. Negative values (cool colours) indicate cooler-than-normal conditions. The black box denotes the study site, Long Lake in the Medicine Bow Mountains of south-eastern, Wyoming. Light grey lines depict lines of latitude/longitude.**

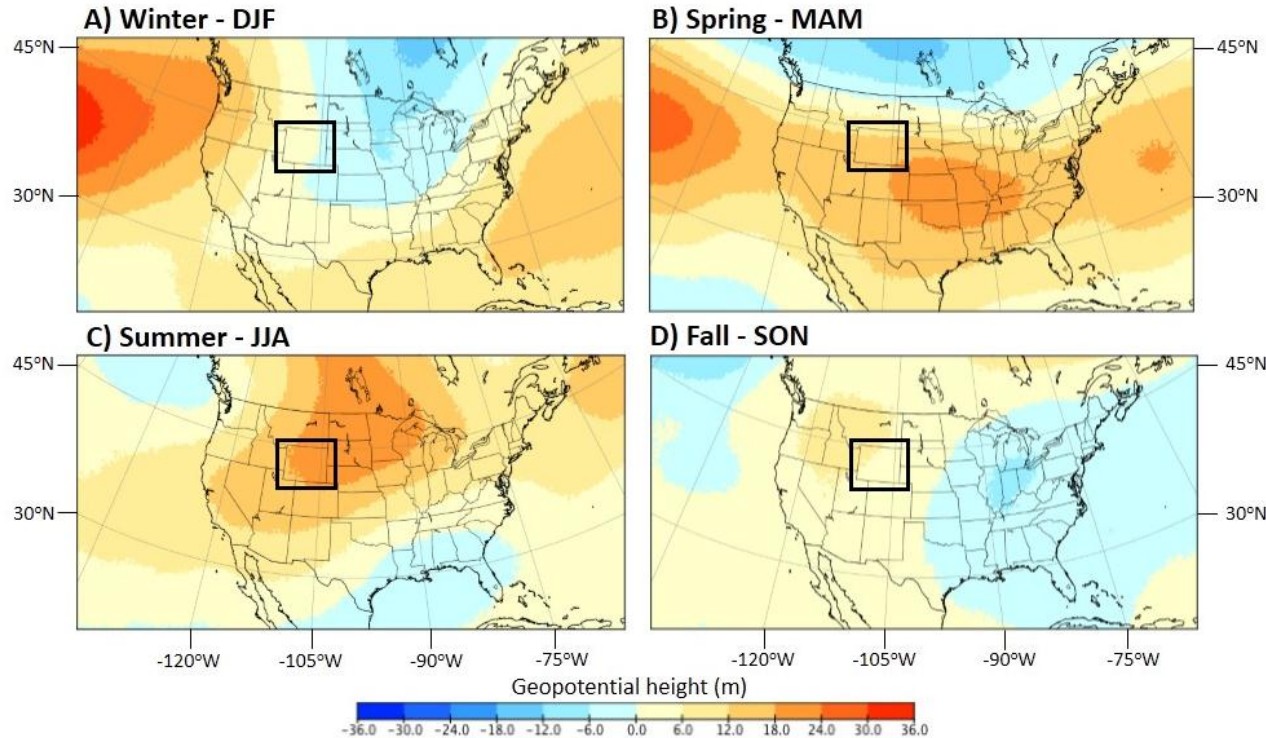

**Figure 5. Composite anomaly maps for 500mb geopotential height during A) the winter season (JJA); B) spring (MAM); C) summer (JJA); and D) fall season (SON). Positive values (warm colours) for 500mb geopotential heights indicate a stronger-than-normal ridge. Negative values (cool colours) indicate a strong-than-normal trough. The black box denotes the study region, Long Lake in the Medicine Bow Mountains of south-eastern, Wyoming. Light grey lines depict lines of latitude/longitude.**

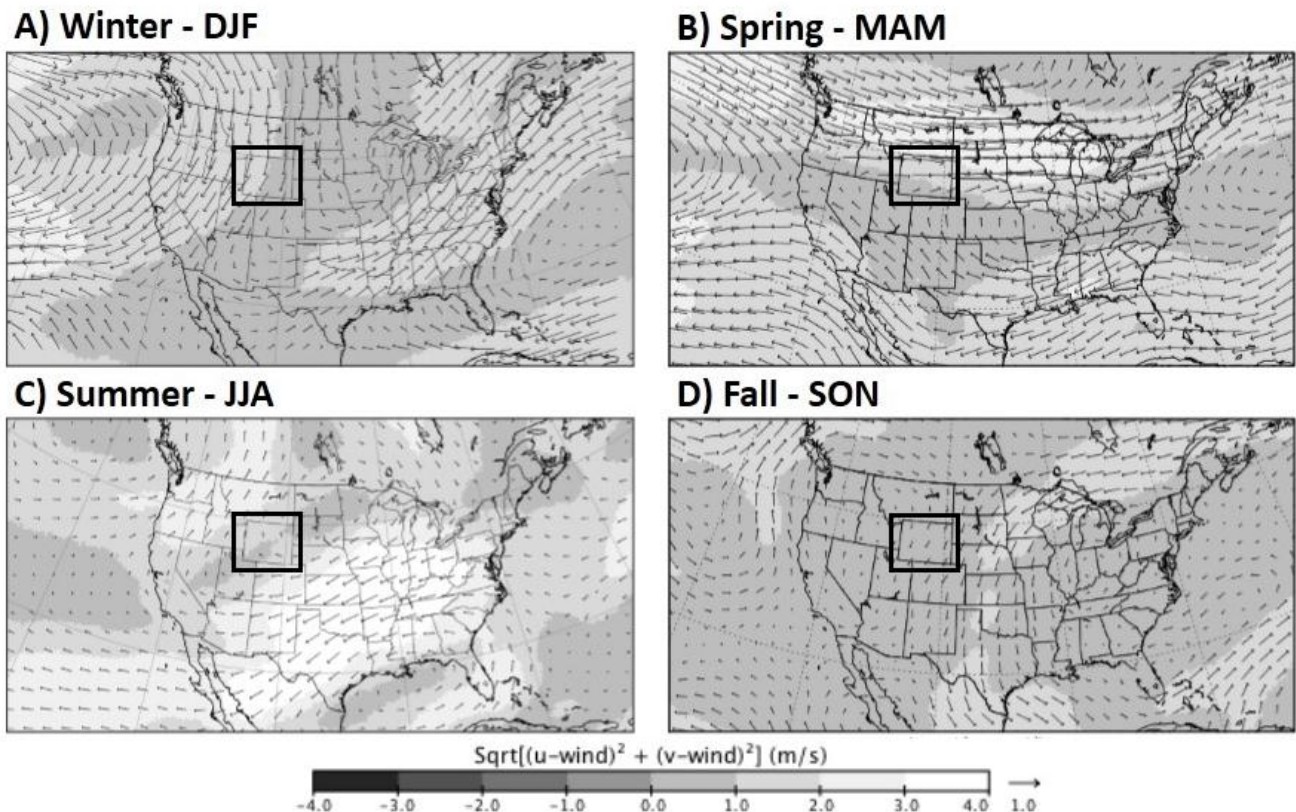

**Figure 6. Seasonal composite anomaly maps for 500mb vector winds during A) the winter season (DJF); B) spring season (MAM); C) summer season (JJA); and D) fall season (SON). The black box denotes the study region, Long Lake in the Medicine Bow Mountains of south-eastern, Wyoming.**

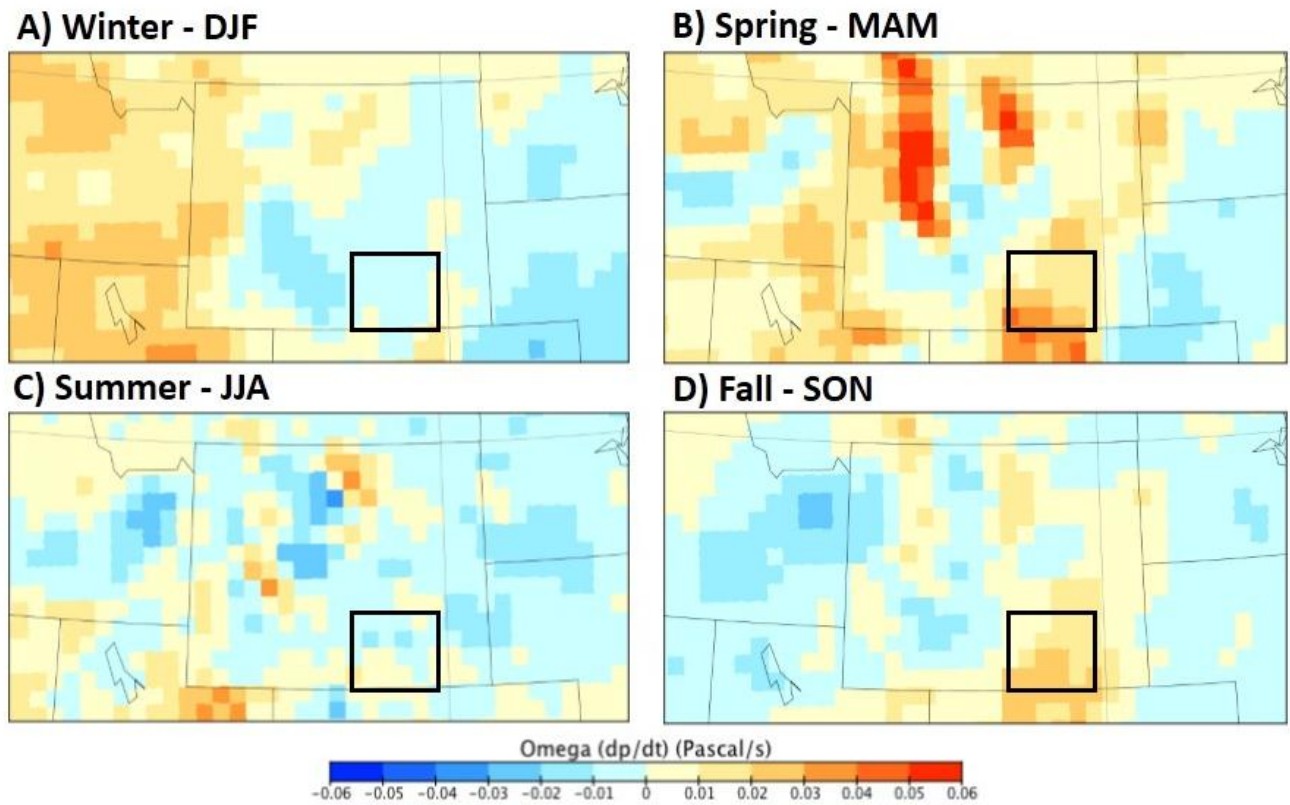

**Figure 7. Composite anomaly maps for 500-mb Omega (vertical velocity) during A) the winter season (DJF); B) spring season (MAM); C) summer season (JJA); and D) the fall season (SON). Positive values (warm colours) for omega indicate enhanced sinking motions (suppress precipitation). Negative values (cool colours) indicate enhanced rising motions (enhanced precipitation). The black box denotes the study site, Long Lake in the Medicine Bow Mountains of south-eastern, Wyoming.**

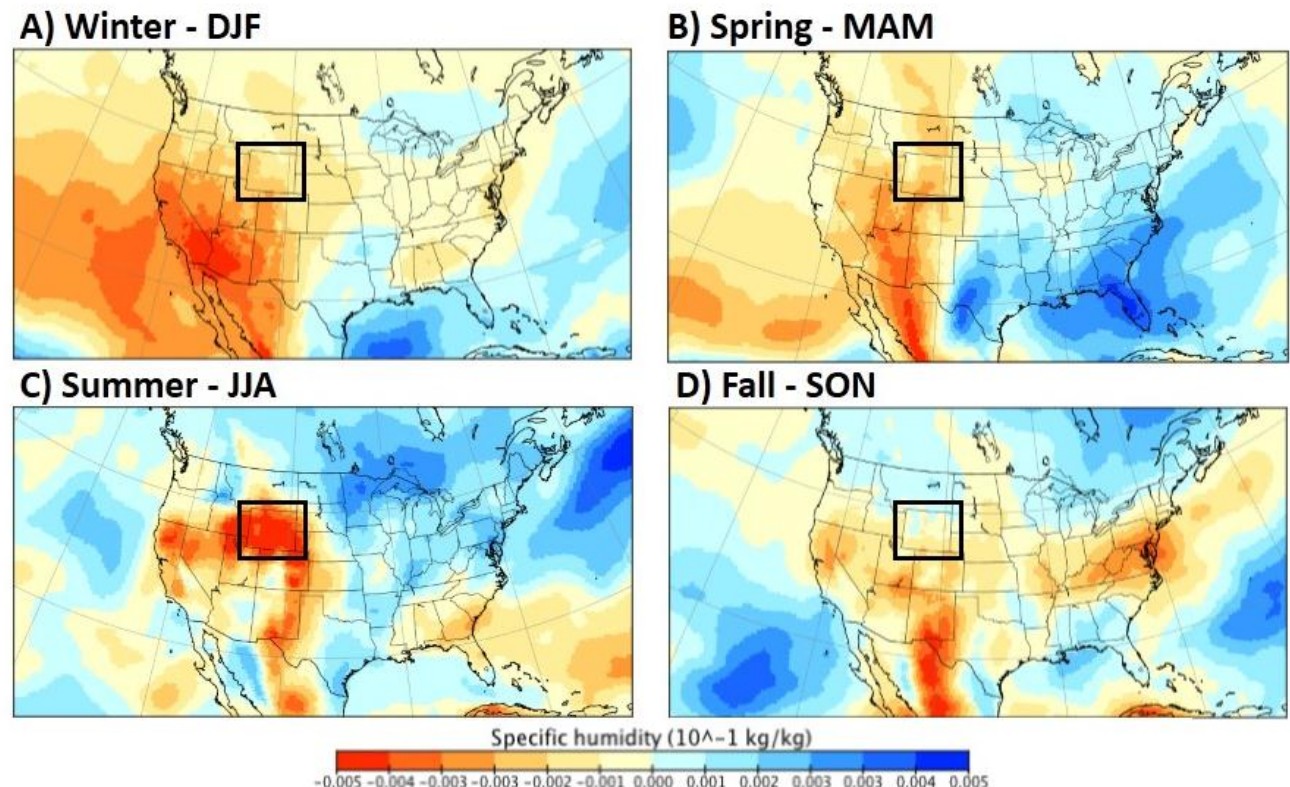

**Figure 8. Composite anomaly maps for 850mb specific humidity during A) the winter season (DJF); B) spring season (MAM); C) summer season (JJA); and D) the fall season (SON). Positive values (cool colours) for 850-mb specific humidity indicate wetter-than-normal conditions in the atmosphere. Negative values (warm colours) indicate dryer-than-normal conditions. The black box denotes the study region, Long Lake in the Medicine Bow Mountains of south-eastern, Wyoming.**

| Climate Variable | Level in the Atmosphere | Purpose of Climate Variable |
|---|---|---|
| Precipitation Rate | Surface | Provides information on how much precipitation makes it to the surface |
| Surface temperature | Surface | Provides information on temperatures at the surface |
| Soil Moisture | Surface | Provides information on surface moisture potentially available for vegetation and the atmosphere. |
| Geopotential Height | 500mb level | Provides information about atmospheric pressure in the mid-troposphere |
| Vector Winds | 500mb level | Provides information about wind direction and anomalous componenet of flow |
| Specific Humidity | 850mb level | Provides information about mositure availablity in the atmosphere |
| Omega (Vertical Velocity) | 500mb level | Provides information about rising or sinking motions in the atmosphere that enhance or suppress precipitation, respectively. |

**Table 1. Climate variables available in the NARR dataset that this particular study used for this analysis.**

