# Peer review of "Drought and vegetation change in the central Rocky Mountains: Potential climatic mechanisms associated with drought conditions at 4200 cal yr BP."

_Climate of the Past, 2017_

## Short Comment (SC1) · 4 Nov 2017

The PAGES Data Stewardship Integrative Activity seeks to advance best practices for sharing the data generated and assembled as part of all PAGES-related activities. The CP Special Issue, "PAGES Young Scientists Meeting 2017" is part of this PAGES activity. The co-editors of the Special Issue are reviewing the data availability within each of the CP-Discussion papers in relation to the CP data policy (https://www.climate-of-the-past.net/about/data_policy.html) and current best practices. The editor team is making recommendations for each paper, with the goal of achieving a high and consistent level

of data stewardship across the Special Issue. We recognize that an additional effort will likely be required to meet the high level of data stewardship envisaged, and we appreciate the dedication and contribution of the authors. This includes the use of Data Citations (see example below). Authors are also strongly encouraged to deposit significant code into a suitable repository and to cite it using a Data Citation.

We ask authors to respond to our comments as part of the regular open interactive discussion. If you have any questions about PAGES Data Stewardship principles, please contact any of us directly. Best wishes for the success of your paper.

YSM Special Issue editor team

H. Plumpton, D.S. Kaufman, R. Barnett, M.F. Loutre, M.N. Evans, S.C. Fritz, C. Tabor, Y. Zhang, E. Razanatsoa, and E. Dearing Crampton Flood

For this paper:

(1) Research input data – surface and atmospheric variables, charcoal, pollen data

This research contribution discusses published proxy data (pollen, charcoal) from Carter et al. (2017), which are already uploaded to long-standing data repositories. However, the URLs provided do not link to the actual data. In order to adhere to the Data Policy for submissions to Climate of the Past, URLs or full data citations to the actual data must be included in the Data Availability section. The data on surface and atmospheric variables used in the composite-anomaly analysis from NOAA also requires citation. A link to a data-viewer web interface is not a persistent identifier of the data behind the interface; a data citation of the actual dataset in a public data repository is required.

(2) Research output data – composite-anomaly values (precipitation, air temperature, 500mb geopotential height, 500mb Omega, 850mb specific humidity, 500mb vector winds)

This paper presents new and valuable composite-anomaly data for numerous climate

variables for Long Lake in the Medicine Bow Mountains of southeastern Wyoming. In order to adhere to the Data Policy for submissions to Climate of the Past, these new data must be uploaded to a long-standing online data repository, and a Data Citation or URL link for access to these data must be provided in the Data Availability section of the paper.

————-

What is a "Data Citation"?

Data Citations track the provenance of a dataset giving credit to the data generator; this is in addition to any references to publications where the data are described. Data Citations are used in the text (or tables) alongside and in the same way as publication citations. In the Reference list, they include: Creators, Title, Repository, Identifier, Submission Year. More information about Data Citations is here: <https://www.datacite.org/mission.html> Here is an example of text and corresponding citations (using CP punctuation style):

"The PAGES2k Consortium (2017a) assembled a large global dataset of temperature-sensitive proxy records (PAGES2k Consortium, 2017b). Among the records is the paleo-temperature reconstruction from Laguna Chepical (de Jong et al., 2016), which was described by de Jong et al. (2013)."

References

de Jong, R., von Gunten, l., Maldonado, A., and Grosjean, M.: Late Holocene summer temperatures in the central Andes reconstructed from the sediments of high-elevation Laguna Chepical, Chile (32° S), Climate of the Past, 9, 1921-1932, 2013.

de Jong, R., von Gunten, l., Maldonado, A., and Grosjean, M.: Laguna Chepical summer temperature reconstruction, World Data Center for Paleoclimatology, https://www.ncdc.noaa.gov/paleo/study/20366, 2016.

PAGES 2k Consortium: A global multiproxy database for temperature reconstructions

of the Common Era, Scientific Data, 4,170088, 2017a.

PAGES 2k Consortium: A global multiproxy database for temperature reconstructions of the Common Era, version 2.0.0, figshare, https://figshare.com/s/d327a0367bb908a4c4f2, 2017b.

---

## Referee Comment (RC1) · Anonymous Referee #1 · 17 Nov 2017

The purpose of the paper is to investigate the mechanisms associated with a mega drought in the Rocky Mountains 4200 years ago. To this end the authors study the atmospheric conditions related to recent years with drought. Five years with drought are identified from a time-series of precipitation anomalies. Composites over these five years of atmospheric fields such as temperature, geopotential height, and winds are then calculated. The features of these composites for different seasons are discussed and used as analogues for the conditions during the mega drought.

I find the subject interesting, but unfortunately the analysis presented in the present

paper is not adequate and convincing. I have basically two major objections with the paper in its present form.

1) The composites are based on only five events. But there is no attempt anywhere in the paper to address the statistical significance or the robustness of the results. The features of the maps may easily in many cases be just results of chance. This should be investigated by calculating and showing the statistical significance. Also, it should be tested if the results are robust and if they depend on one or a few of the five events. It should also be tested if results depend on the threshold (-1.5 standard deviations).

2) The duration of the modern analogues are around a year, while the duration of the mega drought is more than 100 years. Is there any reason at all to believe that events on such different time-scales have the same or related mechanisms? Long lasting events tend, in general, to also be more spatially extended. See e.g. DOI: 10.1002/2016RG000521 for a review of how the number of spatial degrees of freedom depends on the temporal scale considered. The validity of the method of modern analogues should be investigated and discussed in detail.

There is a lot of available model experiments (e.g., CMIP5) where this could be investigated.

Minor comments:

p6, l13: Why are these years "suitable analogues". Are other conditions than the drought index used?

Figure 2: What is the value of the standard deviation?

---

## Author Comment (AC1) · 18 Nov 2017

| Depth | age | deposition time | volume | char counts | char conc. | char influx |
|---|---|---|---|---|---|---|
| 1 | -60 | 3 | 3 | 8 | 2.666666667 | 0.888888889 |
| 2 | -57 | 3 | 5 | 22 | 4.4 | 1.466666667 |
| 3 | -53 | 4 | 5 | 22 | 4.4 | 1.1 |
| 4 | -50 | 3 | 5 | 20 | 4 | 1.333333333 |
| 5 | -46 | 4 | 5 | 26 | 5.2 | 1.3 |
| 6 | -42 | 4 | 5 | 33 | 6.6 | 1.65 |
| 7 | -38 | 4 | 5 | 67 | 13.4 | 3.35 |
| 8 | -33 | 5 | 5 | 207 | 41.4 | 8.28 |
| 9 | -29 | 4 | 5 | 111 | 22.2 | 5.55 |
| 10 | -23 | 6 | 5 | 74 | 14.8 | 2.466666667 |
| 11 | -18 | 5 | 5 | 40 | 8 | 1.6 |
| 12 | -11 | 7 | 5 | 21 | 4.2 | 0.6 |
| 13 | -5 | 6 | 5 | 34 | 6.8 | 1.133333333 |
| 14 | 3 | 8 | 5 | 37 | 7.4 | 0.925 |
| 15 | 11 | 8 | 5 | 49 | 9.8 | 1.225 |
| 16 | 19 | 8 | 5 | 50 | 10 | 1.25 |
| 17 | 28 | 9 | 5 | 36 | 7.2 | 0.8 |
| 18 | 38 | 10 | 5 | 73 | 14.6 | 1.46 |
| 19 | 49 | 11 | 5 | 206 | 41.2 | 3.745454545 |
| 20 | 60 | 11 | 5 | 228 | 45.6 | 4.145454545 |
| 21 | 72 | 12 | 5 | 548 | 109.6 | 9.133333333 |
| 22 | 85 | 13 | 5 | 116 | 23.2 | 1.784615385 |
| 23 | 98 | 13 | 5 | 92 | 18.4 | 1.415384615 |
| 24 | 112 | 14 | 5 | 922 | 184.4 | 13.17142857 |
| 25 | 127 | 15 | 5 | 58 | 11.6 | 0.773333333 |
| 26 | 142 | 15 | 5 | 49 | 9.8 | 0.653333333 |
| 27 | 159 | 17 | 5 | 251 | 50.2 | 2.952941176 |
| 28 | 176 | 17 | 5 | 22 | 4.4 | 0.258823529 |
| 29 | 194 | 18 | 5 | 23 | 4.6 | 0.255555556 |
| 30 | 212 | 18 | 5 | 41 | 8.2 | 0.455555556 |
| 31 | 231 | 19 | 5 | 111 | 22.2 | 1.168421053 |
| 32 | 252 | 21 | 5 | 23 | 4.6 | 0.219047619 |
| 33 | 272 | 20 | 5 | 51 | 10.2 | 0.51 |
| 34 | 294 | 22 | 5 | 73 | 14.6 | 0.663636364 |
| 35 | 315 | 21 | 5 | 65 | 13 | 0.619047619 |
| 36 | 338 | 23 | 5 | 215 | 43 | 1.869565217 |
| 37 | 361 | 23 | 5 | 278 | 55.6 | 2.417391304 |
| 38 | 384 | 23 | 5 | 93 | 18.6 | 0.808695652 |
| 39 | 408 | 24 | 5 | 75 | 15 | 0.625 |
| 40 | 432 | 24 | 5 | 112 | 22.4 | 0.933333333 |
| 41 | 456 | 24 | 5 | 43 | 8.6 | 0.358333333 |
| 42 | 481 | 25 | 5 | 64 | 12.8 | 0.512 |
| 43 | 506 | 25 | 5 | 183 | 36.6 | 1.464 |
| 44 | 531 | 25 | 5 | 110 | 22 | 0.88 |
| 45 | 556 | 25 | 5 | 54 | 10.8 | 0.432 |
| 46 | 581 | 25 | 5 | 25 | 5 | 0.2 |

| 47 | 606 | 25 | 5 | 86 | 17.2 | 0.688 |
|----|------|----|---|------|-------|-------------|
| 48 | 632 | 26 | 5 | 757 | 151.4 | 5.823076923 |
| 49 | 657 | 25 | 5 | 125 | 25 | 1 |
| 50 | 682 | 25 | 5 | 368 | 73.6 | 2.944 |
| 51 | 707 | 25 | 5 | 32 | 6.4 | 0.256 |
| 52 | 732 | 25 | 5 | 30 | 6 | 0.24 |
| 53 | 757 | 25 | 5 | 90 | 18 | 0.72 |
| 54 | 782 | 25 | 5 | 147 | 29.4 | 1.176 |
| 55 | 806 | 24 | 5 | 195 | 39 | 1.625 |
| 56 | 831 | 25 | 5 | 154 | 30.8 | 1.232 |
| 57 | 856 | 25 | 5 | 409 | 81.8 | 3.272 |
| 58 | 882 | 26 | 5 | 259 | 51.8 | 1.992307692 |
| 59 | 907 | 25 | 5 | 1032 | 206.4 | 8.256 |
| 60 | 932 | 25 | 5 | 155 | 31 | 1.24 |
| 61 | 958 | 26 | 5 | 143 | 28.6 | 1.1 |
| 62 | 983 | 25 | 5 | 106 | 21.2 | 0.848 |
| 63 | 1009 | 26 | 5 | 119 | 23.8 | 0.915384615 |
| 64 | 1035 | 26 | 5 | 197 | 39.4 | 1.515384615 |
| 65 | 1061 | 26 | 5 | 258 | 51.6 | 1.984615385 |
| 66 | 1088 | 27 | 5 | 214 | 42.8 | 1.585185185 |
| 67 | 1115 | 27 | 5 | 415 | 83 | 3.074074074 |
| 68 | 1142 | 27 | 5 | 131 | 26.2 | 0.97037037 |
| 69 | 1170 | 28 | 5 | 131 | 26.2 | 0.935714286 |
| 70 | 1197 | 27 | 5 | 63 | 12.6 | 0.466666667 |
| 71 | 1226 | 29 | 5 | 119 | 23.8 | 0.820689655 |
| 72 | 1254 | 28 | 5 | 361 | 72.2 | 2.578571429 |
| 73 | 1283 | 29 | 5 | 111 | 22.2 | 0.765517241 |
| 74 | 1313 | 30 | 5 | 121 | 24.2 | 0.806666667 |
| 75 | 1343 | 30 | 5 | 206 | 41.2 | 1.373333333 |
| 76 | 1373 | 30 | 5 | 106 | 21.2 | 0.706666667 |
| 77 | 1404 | 31 | 5 | 127 | 25.4 | 0.819354839 |
| 78 | 1436 | 32 | 5 | 192 | 38.4 | 1.2 |
| 79 | 1468 | 32 | 5 | 87 | 17.4 | 0.54375 |
| 80 | 1501 | 33 | 5 | 782 | 156.4 | 4.739393939 |
| 81 | 1534 | 33 | 5 | 181 | 36.2 | 1.096969697 |
| 82 | 1568 | 34 | 5 | 456 | 91.2 | 2.682352941 |
| 83 | 1602 | 34 | 5 | 225 | 45 | 1.323529412 |
| 84 | 1637 | 35 | 5 | 103 | 20.6 | 0.588571429 |
| 85 | 1673 | 36 | 5 | 120 | 24 | 0.666666667 |
| 86 | 1710 | 37 | 5 | 108 | 21.6 | 0.583783784 |
| 87 | 1747 | 37 | 5 | 299 | 59.8 | 1.616216216 |
| 88 | 1786 | 39 | 5 | 162 | 32.4 | 0.830769231 |
| 89 | 1825 | 39 | 5 | 499 | 99.8 | 2.558974359 |
| 90 | 1864 | 39 | 5 | 357 | 71.4 | 1.830769231 |
| 91 | 1905 | 41 | 5 | 314 | 62.8 | 1.531707317 |
| 92 | 1946 | 41 | 5 | 72 | 14.4 | 0.351219512 |
| 93 | 1987 | 41 | 5 | 131 | 26.2 | 0.63902439 |

| 94 | 2029 | 42 | 5 | 152 | 30.4 | 0.723809524 |
| 95 | 2072 | 43 | 5 | 304 | 60.8 | 1.413953488 |
| 96 | 2115 | 43 | 5 | 324 | 64.8 | 1.506976744 |
| 97 | 2158 | 43 | 5 | 404 | 80.8 | 1.879069767 |
| 98 | 2202 | 44 | 5 | 129 | 25.8 | 0.586363636 |
| 99 | 2246 | 44 | 5 | 404 | 80.8 | 1.836363636 |
| 100 | 2290 | 44 | 5 | 294 | 58.8 | 1.336363636 |
| 101 | 2334 | 44 | 5 | 93 | 18.6 | 0.422727273 |
| 102 | 2379 | 45 | 5 | 286 | 57.2 | 1.271111111 |
| 103 | 2423 | 44 | 5 | 147 | 29.4 | 0.668181818 |
| 104 | 2468 | 45 | 5 | 264 | 52.8 | 1.173333333 |
| 105 | 2512 | 44 | 5 | 147 | 29.4 | 0.668181818 |
| 106 | 2557 | 45 | 5 | 112 | 22.4 | 0.497777778 |
| 107 | 2601 | 44 | 5 | 69 | 13.8 | 0.313636364 |
| 108 | 2645 | 44 | 5 | 219 | 43.8 | 0.995454545 |
| 109 | 2689 | 44 | 5 | 499 | 99.8 | 2.268181818 |
| 110 | 2732 | 43 | 5 | 98 | 19.6 | 0.455813953 |
| 111 | 2775 | 43 | 5 | 291 | 58.2 | 1.353488372 |
| 112 | 2818 | 43 | 5 | 45 | 9 | 0.209302326 |
| 113 | 2860 | 42 | 5 | 77 | 15.4 | 0.366666667 |
| 114 | 2902 | 42 | 5 | 173 | 34.6 | 0.823809524 |
| 115 | 2943 | 41 | 5 | 110 | 22 | 0.536585366 |
| 116 | 2984 | 41 | 5 | 84 | 16.8 | 0.409756098 |
| 117 | 3024 | 40 | 5 | 647 | 129.4 | 3.235 |
| 118 | 3063 | 39 | 5 | 53 | 10.6 | 0.271794872 |
| 119 | 3102 | 39 | 5 | 369 | 73.8 | 1.892307692 |
| 120 | 3139 | 37 | 5 | 113 | 22.6 | 0.610810811 |
| 121 | 3176 | 37 | 5 | 204 | 40.8 | 1.102702703 |
| 122 | 3212 | 36 | 5 | 353 | 70.6 | 1.961111111 |
| 123 | 3247 | 35 | 5 | 61 | 12.2 | 0.348571429 |
| 124 | 3280 | 33 | 5 | 133 | 26.6 | 0.806060606 |
| 125 | 3313 | 33 | 5 | 138 | 27.6 | 0.836363636 |
| 126 | 3345 | 32 | 5 | 254 | 50.8 | 1.5875 |
| 127 | 3375 | 30 | 5 | 866 | 173.2 | 5.773333333 |
| 128 | 3404 | 29 | 5 | 228 | 45.6 | 1.572413793 |
| 129 | 3432 | 28 | 5 | 166 | 33.2 | 1.185714286 |
| 130 | 3459 | 27 | 5 | 172 | 34.4 | 1.274074074 |
| 131 | 3484 | 25 | 5 | 354 | 70.8 | 2.832 |
| 132 | 3508 | 24 | 5 | 392 | 78.4 | 3.266666667 |
| 133 | 3531 | 23 | 5 | 273 | 54.6 | 2.373913043 |
| 134 | 3553 | 22 | 5 | 105 | 21 | 0.954545455 |
| 135 | 3574 | 21 | 5 | 109 | 21.8 | 1.038095238 |
| 136 | 3594 | 20 | 5 | 120 | 24 | 1.2 |
| 137 | 3614 | 20 | 5 | 578 | 115.6 | 5.78 |
| 138 | 3632 | 18 | 5 | 957 | 191.4 | 10.63333333 |
| 139 | 3651 | 19 | 5 | 343 | 68.6 | 3.610526316 |
| 140 | 3668 | 17 | 5 | 126 | 25.2 | 1.482352941 |

| 141 | 3685 | 17 | 5 | 113 | 22.6 | 1.329411765 |
| 142 | 3702 | 17 | 5 | 104 | 20.8 | 1.223529412 |
| 143 | 3718 | 16 | 5 | 79 | 15.8 | 0.9875 |
| 144 | 3735 | 17 | 5 | 160 | 32 | 1.882352941 |
| 145 | 3751 | 16 | 5 | 949 | 189.8 | 11.8625 |
| 146 | 3767 | 16 | 5 | 172 | 34.4 | 2.15 |
| 147 | 3783 | 16 | 5 | 104 | 20.8 | 1.3 |
| 148 | 3799 | 16 | 5 | 79 | 15.8 | 0.9875 |
| 149 | 3815 | 16 | 5 | 280 | 56 | 3.5 |
| 150 | 3832 | 17 | 5 | 871 | 174.2 | 10.24705882 |
| 151 | 3849 | 17 | 5 | 67 | 13.4 | 0.788235294 |
| 152 | 3866 | 17 | 5 | 128 | 25.6 | 1.505882353 |
| 153 | 3884 | 18 | 5 | 158 | 31.6 | 1.755555556 |
| 154 | 3902 | 18 | 5 | 189 | 37.8 | 2.1 |
| 155 | 3921 | 19 | 5 | 567 | 113.4 | 5.968421053 |
| 156 | 3941 | 20 | 5 | 140 | 28 | 1.4 |
| 157 | 3962 | 21 | 5 | 94 | 18.8 | 0.895238095 |
| 158 | 3983 | 21 | 5 | 253 | 50.6 | 2.40952381 |
| 159 | 4006 | 23 | 5 | 310 | 62 | 2.695652174 |
| 160 | 4029 | 23 | 5 | 229 | 45.8 | 1.991304348 |
| 161 | 4053 | 24 | 5 | 710 | 142 | 5.916666667 |
| 162 | 4077 | 24 | 5 | 109 | 21.8 | 0.908333333 |
| 163 | 4103 | 26 | 5 | 267 | 53.4 | 2.053846154 |
| 164 | 4129 | 26 | 5 | 262 | 52.4 | 2.015384615 |
| 165 | 4156 | 27 | 5 | 225 | 45 | 1.666666667 |
| 166 | 4184 | 28 | 5 | 62 | 12.4 | 0.442857143 |
| 167 | 4213 | 29 | 5 | 148 | 29.6 | 1.020689655 |
| 168 | 4242 | 29 | 5 | 80 | 16 | 0.551724138 |
| 169 | 4272 | 30 | 5 | 70 | 14 | 0.466666667 |
| 170 | 4303 | 31 | 5 | 239 | 47.8 | 1.541935484 |
| 171 | 4335 | 32 | 5 | 135 | 27 | 0.84375 |
| 172 | 4368 | 33 | 5 | 124 | 24.8 | 0.751515152 |
| 173 | 4401 | 33 | 5 | 722 | 144.4 | 4.375757576 |
| 174 | 4435 | 34 | 5 | 39 | 7.8 | 0.229411765 |
| 175 | 4470 | 35 | 5 | 55 | 11 | 0.314285714 |
| 176 | 4505 | 35 | 5 | 376 | 75.2 | 2.148571429 |
| 177 | 4541 | 36 | 5 | 362 | 72.4 | 2.011111111 |
| 178 | 4578 | 37 | 5 | 284 | 56.8 | 1.535135135 |
| 179 | 4616 | 38 | 5 | 137 | 27.4 | 0.721052632 |
| 180 | 4655 | 39 | 5 | 413 | 82.6 | 2.117948718 |
| 181 | 4694 | 39 | 5 | 68 | 13.6 | 0.348717949 |
| 182 | 4734 | 40 | 5 | 59 | 11.8 | 0.295 |
| 183 | 4774 | 40 | 5 | 204 | 40.8 | 1.02 |
| 184 | 4816 | 42 | 5 | 180 | 36 | 0.857142857 |
| 185 | 4858 | 42 | 5 | 80 | 16 | 0.380952381 |
| 186 | 4900 | 42 | 5 | 197 | 39.4 | 0.938095238 |
| 187 | 4944 | 44 | 5 | 132 | 26.4 | 0.6 |

| | | | | | | |
|-----|------|----|---|-----|-------|-------------|
| 188 | 4988 | 44 | 5 | 398 | 79.6  | 1.809090909 |
| 189 | 5033 | 45 | 5 | 31  | 6.2   | 0.137777778 |
| 190 | 5078 | 45 | 5 | 43  | 8.6   | 0.191111111 |
| 191 | 5124 | 46 | 5 | 88  | 17.6  | 0.382608696 |
| 192 | 5171 | 47 | 5 | 109 | 21.8  | 0.463829787 |
| 193 | 5218 | 47 | 5 | 78  | 15.6  | 0.331914894 |
| 194 | 5267 | 49 | 5 | 626 | 125.2 | 2.555102041 |
| 195 | 5315 | 48 | 5 | 570 | 114   | 2.375       |
| 196 | 5365 | 50 | 5 | 46  | 9.2   | 0.184       |
| 197 | 5415 | 50 | 5 | 23  | 4.6   | 0.092       |
| 198 | 5466 | 51 | 5 | 136 | 27.2  | 0.533333333 |
| 199 | 5517 | 51 | 5 | 7   | 1.4   | 0.02745098  |
| 200 | 5569 | 52 | 5 | 16  | 3.2   | 0.061538462 |
| 201 | 5622 | 53 | 5 | 76  | 15.2  | 0.286792453 |
| 202 | 5675 | 53 | 5 | 335 | 67    | 1.264150943 |
| 203 | 5729 | 54 | 5 | 56  | 11.2  | 0.207407407 |
| 204 | 5784 | 55 | 5 | 105 | 21    | 0.381818182 |
| 205 | 5839 | 55 | 5 | 539 | 107.8 | 1.96        |
| 206 | 5894 | 55 | 5 | 150 | 30    | 0.545454545 |
| 207 | 5951 | 57 | 5 | 168 | 33.6  | 0.589473684 |
| 208 | 6008 | 57 | 5 | 111 | 22.2  | 0.389473684 |
| 209 | 6065 | 57 | 5 | 241 | 48.2  | 0.845614035 |
| 210 | 6124 | 59 | 5 | 273 | 54.6  | 0.925423729 |
| 211 | 6182 | 58 | 5 | 110 | 22    | 0.379310345 |
| 212 | 6242 | 60 | 5 | 200 | 40    | 0.666666667 |
| 213 | 6302 | 60 | 5 | 38  | 7.6   | 0.126666667 |
| 214 | 6362 | 60 | 5 | 177 | 35.4  | 0.59        |
| 215 | 6423 | 61 | 5 | 150 | 30    | 0.491803279 |
| 216 | 6485 | 62 | 5 | 53  | 10.6  | 0.170967742 |
| 217 | 6547 | 62 | 5 | 195 | 39    | 0.629032258 |
| 218 | 6610 | 63 | 5 | 108 | 21.6  | 0.342857143 |
| 219 | 6673 | 63 | 5 | 297 | 59.4  | 0.942857143 |
| 220 | 6737 | 64 | 5 | 47  | 9.4   | 0.146875    |
| 221 | 6802 | 65 | 5 | 40  | 8     | 0.123076923 |
| 222 | 6867 | 65 | 5 | 436 | 87.2  | 1.341538462 |
| 223 | 6932 | 65 | 5 | 191 | 38.2  | 0.587692308 |
| 224 | 6998 | 66 | 5 | 114 | 22.8  | 0.345454545 |
| 225 | 7065 | 67 | 5 | 192 | 38.4  | 0.573134328 |
| 226 | 7132 | 67 | 5 | 206 | 41.2  | 0.614925373 |
| 227 | 7200 | 68 | 5 | 117 | 23.4  | 0.344117647 |
| 228 | 7268 | 68 | 5 | 103 | 20.6  | 0.302941176 |
| 229 | 7337 | 69 | 5 | 145 | 29    | 0.420289855 |
| 230 | 7406 | 69 | 5 | 249 | 49.8  | 0.72173913  |
| 231 | 7475 | 69 | 5 | 178 | 35.6  | 0.515942029 |
| 232 | 7546 | 71 | 5 | 216 | 43.2  | 0.608450704 |
| 233 | 7616 | 70 | 5 | 122 | 24.4  | 0.348571429 |
| 234 | 7687 | 71 | 5 | 246 | 49.2  | 0.692957746 |

| | | | | | | |
|---|---|---|---|---|---|---|
| 235 | 7759 | 72 | 5 | 127 | 25.4 | 0.352777778 |
| 236 | 7830 | 71 | 5 | 251 | 50.2 | 0.707042254 |
| 237 | 7902 | 72 | 5 | 401 | 80.2 | 1.113888889 |
| 238 | 7973 | 71 | 5 | 200 | 40 | 0.563380282 |
| 239 | 8045 | 72 | 5 | 217 | 43.4 | 0.602777778 |
| 240 | 8116 | 71 | 5 | 650 | 130 | 1.830985915 |
| 241 | 8187 | 71 | 5 | 427 | 85.4 | 1.202816901 |
| 242 | 8257 | 70 | 5 | 58 | 11.6 | 0.165714286 |
| 243 | 8327 | 70 | 5 | 177 | 35.4 | 0.505714286 |
| 244 | 8396 | 69 | 5 | 65 | 13 | 0.188405797 |
| 245 | 8465 | 69 | 5 | 64 | 12.8 | 0.185507246 |
| 246 | 8533 | 68 | 5 | 34 | 6.8 | 0.1 |
| 247 | 8599 | 66 | 5 | 124 | 24.8 | 0.375757576 |
| 248 | 8665 | 66 | 5 | 131 | 26.2 | 0.396969697 |
| 249 | 8729 | 64 | 5 | 109 | 21.8 | 0.340625 |
| 250 | 8792 | 63 | 5 | 156 | 31.2 | 0.495238095 |
| 251 | 8854 | 62 | 5 | 149 | 29.8 | 0.480645161 |
| 252 | 8915 | 61 | 5 | 163 | 32.6 | 0.53442623 |
| 253 | 8974 | 59 | 5 | 54 | 10.8 | 0.183050847 |
| 254 | 9033 | 59 | 5 | 49 | 9.8 | 0.166101695 |
| 255 | 9090 | 57 | 5 | 146 | 29.2 | 0.512280702 |
| 256 | 9146 | 56 | 5 | 59 | 11.8 | 0.210714286 |
| 257 | 9200 | 54 | 5 | 118 | 23.6 | 0.437037037 |
| 258 | 9254 | 54 | 5 | 141 | 28.2 | 0.522222222 |
| 259 | 9306 | 52 | 5 | 112 | 22.4 | 0.430769231 |
| 260 | 9358 | 52 | 5 | 58 | 11.6 | 0.223076923 |
| 261 | 9408 | 50 | 5 | 74 | 14.8 | 0.296 |
| 262 | 9457 | 49 | 5 | 89 | 17.8 | 0.363265306 |
| 263 | 9505 | 48 | 5 | 142 | 28.4 | 0.591666667 |
| 264 | 9552 | 47 | 5 | 168 | 33.6 | 0.714893617 |
| 265 | 9598 | 46 | 5 | 136 | 27.2 | 0.591304348 |
| 266 | 9643 | 45 | 5 | 121 | 24.2 | 0.537777778 |
| 267 | 9687 | 44 | 5 | 50 | 10 | 0.227272727 |
| 268 | 9730 | 43 | 5 | 66 | 13.2 | 0.306976744 |
| 269 | 9772 | 42 | 5 | 69 | 13.8 | 0.328571429 |
| 270 | 9813 | 41 | 5 | 85 | 17 | 0.414634146 |
| 271 | 9853 | 40 | 5 | 99 | 19.8 | 0.495 |
| 272 | 9892 | 39 | 5 | 107 | 21.4 | 0.548717949 |
| 273 | 9931 | 39 | 5 | 179 | 35.8 | 0.917948718 |
| 274 | 9968 | 37 | 5 | 152 | 30.4 | 0.821621622 |
| 275 | 10004 | 36 | 5 | 136 | 27.2 | 0.755555556 |
| 276 | 10040 | 36 | 5 | 126 | 25.2 | 0.7 |
| 277 | 10075 | 35 | 5 | 68 | 13.6 | 0.388571429 |
| 278 | 10109 | 34 | 5 | 87 | 17.4 | 0.511764706 |
| 279 | 10142 | 33 | 5 | 92 | 18.4 | 0.557575758 |
| 280 | 10175 | 33 | 5 | 72 | 14.4 | 0.436363636 |
| 281 | 10206 | 31 | 5 | 66 | 13.2 | 0.425806452 |

| | | | | | | |
|---|---|---|---|---|---|---|
| 282 | 10237 | 31 | 5 | 95 | 19 | 0.612903226 |
| 283 | 10267 | 30 | 5 | 107 | 21.4 | 0.713333333 |
| 284 | 10297 | 30 | 5 | 121 | 24.2 | 0.806666667 |
| 285 | 10325 | 28 | 5 | 168 | 33.6 | 1.2 |
| 286 | 10353 | 28 | 5 | 81 | 16.2 | 0.578571429 |
| 287 | 10380 | 27 | 5 | 83 | 16.6 | 0.614814815 |
| 288 | 10407 | 27 | 5 | 169 | 33.8 | 1.251851852 |
| 289 | 10433 | 26 | 5 | 302 | 60.4 | 2.323076923 |
| 290 | 10458 | 25 | 5 | 172 | 34.4 | 1.376 |
| 291 | 10483 | 25 | 5 | 25 | 5 | 0.2 |
| 292 | 10507 | 24 | 5 | 37 | 7.4 | 0.308333333 |
| 293 | 10531 | 24 | 5 | 19 | 3.8 | 0.158333333 |
| 294 | 10554 | 23 | 5 | 34 | 6.8 | 0.295652174 |
| 295 | 10576 | 22 | 5 | 44 | 8.8 | 0.4 |
| 296 | 10598 | 22 | 5 | 56 | 11.2 | 0.509090909 |
| 297 | 10620 | 22 | 5 | 34 | 6.8 | 0.309090909 |
| 298 | 10640 | 20 | 5 | 64 | 12.8 | 0.64 |
| 299 | 10661 | 21 | 5 | 59 | 11.8 | 0.561904762 |
| 300 | 10681 | 20 | 5 | 88 | 17.6 | 0.88 |
| 301 | 10700 | 19 | 5 | 80 | 16 | 0.842105263 |
| 302 | 10719 | 19 | 5 | 120 | 24 | 1.263157895 |
| 303 | 10738 | 19 | 5 | 96 | 19.2 | 1.010526316 |
| 304 | 10756 | 18 | 5 | 62 | 12.4 | 0.688888889 |
| 305 | 10774 | 18 | 5 | 47 | 9.4 | 0.522222222 |
| 306 | 10791 | 17 | 5 | 51 | 10.2 | 0.6 |
| 307 | 10808 | 17 | 5 | 97 | 19.4 | 1.141176471 |
| 308 | 10825 | 17 | 5 | 99 | 19.8 | 1.164705882 |
| 309 | 10841 | 16 | 5 | 103 | 20.6 | 1.2875 |
| 310 | 10858 | 17 | 5 | 110 | 22 | 1.294117647 |
| 311 | 10873 | 15 | 5 | 71 | 14.2 | 0.946666667 |
| 312 | 10889 | 16 | 5 | 67 | 13.4 | 0.8375 |
| 313 | 10904 | 15 | 5 | 64 | 12.8 | 0.853333333 |
| 314 | 10919 | 15 | 5 | 74 | 14.8 | 0.986666667 |
| 315 | 10934 | 15 | 5 | 52 | 10.4 | 0.693333333 |
| 316 | 10948 | 14 | 5 | 40 | 8 | 0.571428571 |
| 317 | 10963 | 15 | 5 | 47 | 9.4 | 0.626666667 |
| 318 | 10977 | 14 | 5 | 73 | 14.6 | 1.042857143 |
| 319 | 10991 | 14 | 5 | 75 | 15 | 1.071428571 |
| 320 | 11005 | 14 | 5 | 83 | 16.6 | 1.185714286 |
| 321 | 11019 | 14 | 5 | 95 | 19 | 1.357142857 |
| 322 | 11032 | 13 | 5 | 134 | 26.8 | 2.061538462 |
| 323 | 11046 | 14 | 5 | 155 | 31 | 2.214285714 |
| 324 | 11059 | 13 | 5 | 55 | 11 | 0.846153846 |
| 325 | 11072 | 13 | 5 | 57 | 11.4 | 0.876923077 |
| 326 | 11086 | 14 | 5 | 53 | 10.6 | 0.757142857 |
| 327 | 11099 | 13 | 5 | 79 | 15.8 | 1.215384615 |
| 328 | 11112 | 13 | 5 | 79 | 15.8 | 1.215384615 |

| 329 | 11125 | 13 | 5 | 65 | 13 | 1 |
|-----|-------|----|---|-----|------|-------------|
| 330 | 11138 | 13 | 5 | 82 | 16.4 | 1.261538462 |
| 331 | 11151 | 13 | 5 | 71 | 14.2 | 1.092307692 |
| 332 | 11163 | 12 | 5 | 84 | 16.8 | 1.4 |
| 333 | 11176 | 13 | 5 | 61 | 12.2 | 0.938461538 |
| 334 | 11189 | 13 | 5 | 79 | 15.8 | 1.215384615 |
| 335 | 11201 | 12 | 5 | 114 | 22.8 | 1.9 |
| 336 | 11214 | 13 | 5 | 100 | 20 | 1.538461538 |
| 337 | 11226 | 12 | 5 | 110 | 22 | 1.833333333 |
| 338 | 11238 | 12 | 5 | 101 | 20.2 | 1.683333333 |
| 339 | 11250 | 12 | 5 | 98 | 19.6 | 1.633333333 |
| 340 | 11263 | 13 | 5 | 135 | 27 | 2.076923077 |
| 341 | 11275 | 12 | 5 | 114 | 22.8 | 1.9 |
| 342 | 11287 | 12 | 5 | 144 | 28.8 | 2.4 |
| 343 | 11299 | 12 | 5 | 165 | 33 | 2.75 |
| 344 | 11310 | 11 | 5 | 141 | 28.2 | 2.563636364 |
| 345 | 11322 | 12 | 5 | 88 | 17.6 | 1.466666667 |
| 346 | 11334 | 12 | 5 | 77 | 15.4 | 1.283333333 |
| 347 | 11345 | 11 | 5 | 89 | 17.8 | 1.618181818 |
| 348 | 11357 | 12 | 5 | 53 | 10.6 | 0.883333333 |
| 349 | 11369 | 12 | 5 | 55 | 11 | 0.916666667 |
| 350 | 11380 | 11 | 5 | 49 | 9.8 | 0.890909091 |
| 351 | 11391 | 11 | 5 | 83 | 16.6 | 1.509090909 |
| 352 | 11403 | 12 | 5 | 58 | 11.6 | 0.966666667 |
| 353 | 11414 | 11 | 5 | 48 | 9.6 | 0.872727273 |
| 354 | 11425 | 11 | 5 | 66 | 13.2 | 1.2 |
| 355 | 11436 | 11 | 5 | 37 | 7.4 | 0.672727273 |
| 356 | 11447 | 11 | 5 | 60 | 12 | 1.090909091 |
| 357 | 11458 | 11 | 5 | 92 | 18.4 | 1.672727273 |
| 358 | 11469 | 11 | 5 | 71 | 14.2 | 1.290909091 |
| 359 | 11480 | 11 | 5 | 104 | 20.8 | 1.890909091 |
| 360 | 11490 | 10 | 5 | 97 | 19.4 | 1.94 |
| 361 | 11501 | 11 | 5 | 111 | 22.2 | 2.018181818 |
| 362 | 11512 | 11 | 5 | 115 | 23 | 2.090909091 |
| 363 | 11522 | 10 | 5 | 172 | 34.4 | 3.44 |
| 364 | 11533 | 11 | 5 | 62 | 12.4 | 1.127272727 |
| 365 | 11543 | 10 | 5 | 48 | 9.6 | 0.96 |
| 366 | 11554 | 11 | 5 | 18 | 3.6 | 0.327272727 |
| 367 | 11564 | 10 | 5 | 35 | 7 | 0.7 |
| 368 | 11574 | 10 | 5 | 29 | 5.8 | 0.58 |
| 369 | 11585 | 11 | 5 | 20 | 4 | 0.363636364 |
| 370 | 11595 | 10 | 5 | 32 | 6.4 | 0.64 |
| 371 | 11605 | 10 | 5 | 51 | 10.2 | 1.02 |
| 372 | 11615 | 10 | 5 | 29 | 5.8 | 0.58 |
| 373 | 11625 | 10 | 5 | 79 | 15.8 | 1.58 |
| 374 | 11635 | 10 | 5 | 93 | 18.6 | 1.86 |
| 375 | 11645 | 10 | 5 | 126 | 25.2 | 2.52 |

| 376 | 11655 | 10 | 5 | 81 | 16.2 | 1.62 |
|-----|-------|----|---|-----|------|------|
| 377 | 11665 | 10 | 5 | 278 | 55.6 | 5.56 |
| 378 | 11675 | 10 | 5 | 35 | 7 | 0.7 |
| 379 | 11684 | 9 | 5 | 30 | 6 | 0.666666667 |
| 380 | 11694 | 10 | 5 | 50 | 10 | 1 |
| 381 | 11704 | 10 | 5 | 42 | 8.4 | 0.84 |
| 382 | 11713 | 9 | 5 | 36 | 7.2 | 0.8 |
| 383 | 11723 | 10 | 5 | 59 | 11.8 | 1.18 |
| 384 | 11732 | 9 | 5 | 63 | 12.6 | 1.4 |
| 385 | 11742 | 10 | 5 | 60 | 12 | 1.2 |
| 386 | 11751 | 9 | 5 | 108 | 21.6 | 2.4 |
| 387 | 11760 | 9 | 5 | 85 | 17 | 1.888888889 |
| 388 | 11770 | 10 | 5 | 56 | 11.2 | 1.12 |
| 389 | 11779 | 9 | 5 | 96 | 19.2 | 2.133333333 |
| 390 | 11788 | 9 | 5 | 95 | 19 | 2.111111111 |
| 391 | 11797 | 9 | 5 | 108 | 21.6 | 2.4 |
| 392 | 11807 | 10 | 5 | 100 | 20 | 2 |
| 393 | 11816 | 9 | 5 | 109 | 21.8 | 2.422222222 |
| 394 | 11825 | 9 | 5 | 90 | 18 | 2 |
| 395 | 11834 | 9 | 5 | 75 | 15 | 1.666666667 |
| 396 | 11843 | 9 | 5 | 65 | 13 | 1.444444444 |
| 397 | 11852 | 9 | 5 | 58 | 11.6 | 1.288888889 |
| 398 | 11861 | 9 | 5 | 60 | 12 | 1.333333333 |
| 399 | 11870 | 9 | 5 | 71 | 14.2 | 1.577777778 |
| 400 | 11879 | 9 | 5 | 58 | 11.6 | 1.288888889 |
| 401 | 11887 | 8 | 5 | 77 | 15.4 | 1.925 |
| 402 | 11896 | 9 | 5 | 62 | 12.4 | 1.377777778 |
| 403 | 11905 | 9 | 5 | 78 | 15.6 | 1.733333333 |
| 404 | 11914 | 9 | 5 | 71 | 14.2 | 1.577777778 |
| 405 | 11922 | 8 | 5 | 68 | 13.6 | 1.7 |
| 406 | 11931 | 9 | 5 | 100 | 20 | 2.222222222 |
| 407 | 11940 | 9 | 5 | 73 | 14.6 | 1.622222222 |
| 408 | 11948 | 8 | 5 | 59 | 11.8 | 1.475 |
| 409 | 11957 | 9 | 5 | 66 | 13.2 | 1.466666667 |
| 410 | 11965 | 8 | 5 | 65 | 13 | 1.625 |
| 411 | 11974 | 9 | 5 | 87 | 17.4 | 1.933333333 |
| 412 | 11982 | 8 | 5 | 150 | 30 | 3.75 |
| 413 | 11991 | 9 | 5 | 66 | 13.2 | 1.466666667 |
| 414 | 11999 | 8 | 5 | 43 | 8.6 | 1.075 |
| 415 | 12008 | 9 | 5 | 38 | 7.6 | 0.844444444 |
| 416 | 12016 | 8 | 5 | 75 | 15 | 1.875 |
| 417 | 12025 | 9 | 5 | 91 | 18.2 | 2.022222222 |
| 418 | 12033 | 8 | 5 | 63 | 12.6 | 1.575 |
| 419 | 12041 | 8 | 5 | 44 | 8.8 | 1.1 |
| 420 | 12050 | 9 | 5 | 53 | 10.6 | 1.177777778 |
| 421 | 12058 | 8 | 5 | 38 | 7.6 | 0.95 |
| 422 | 12066 | 8 | 5 | 69 | 13.8 | 1.725 |

| | | | | | | |
|---|---|---|---|---|---|---|
| 423 | 12075 | 9 | 5 | 55 | 11 | 1.222222222 |
| 424 | 12083 | 8 | 5 | 69 | 13.8 | 1.725 |
| 425 | 12091 | 8 | 5 | 42 | 8.4 | 1.05 |
| 426 | 12099 | 8 | 5 | 51 | 10.2 | 1.275 |
| 427 | 12107 | 8 | 5 | 35 | 7 | 0.875 |
| 428 | 12116 | 9 | 5 | 19 | 3.8 | 0.422222222 |
| 429 | 12124 | 8 | 5 | 20 | 4 | 0.5 |
| 430 | 12132 | 8 | 5 | 28 | 5.6 | 0.7 |
| 431 | 12140 | 8 | 5 | 34 | 6.8 | 0.85 |
| 432 | 12148 | 8 | 5 | 41 | 8.2 | 1.025 |
| 433 | 12156 | 8 | 5 | 52 | 10.4 | 1.3 |
| 434 | 12164 | 8 | 5 | 37 | 7.4 | 0.925 |
| 435 | 12172 | 8 | 5 | 36 | 7.2 | 0.9 |
| 436 | 12181 | 9 | 5 | 60 | 12 | 1.333333333 |
| 437 | 12189 | 8 | 5 | 70 | 14 | 1.75 |
| 438 | 12197 | 8 | 5 | 67 | 13.4 | 1.675 |
| 439 | 12205 | 8 | 5 | 113 | 22.6 | 2.825 |
| 440 | 12213 | 8 | 5 | 105 | 21 | 2.625 |
| 441 | 12221 | 8 | 5 | 33 | 6.6 | 0.825 |
| 442 | 12229 | 8 | 5 | 23 | 4.6 | 0.575 |
| 443 | 12237 | 8 | 5 | 31 | 6.2 | 0.775 |
| 444 | 12245 | 8 | 5 | 28 | 5.6 | 0.7 |
| 445 | 12253 | 8 | 5 | 19 | 3.8 | 0.475 |
| 446 | 12261 | 8 | 5 | 27 | 5.4 | 0.675 |
| 447 | 12269 | 8 | 5 | 36 | 7.2 | 0.9 |
| 448 | 12277 | 8 | 5 | 37 | 7.4 | 0.925 |
| 449 | 12285 | 8 | 5 | 31 | 6.2 | 0.775 |
| 450 | 12293 | 8 | 5 | 17 | 3.4 | 0.425 |
| 451 | 12301 | 8 | 5 | 27 | 5.4 | 0.675 |
| 452 | 12309 | 8 | 5 | 32 | 6.4 | 0.8 |
| 453 | 12317 | 8 | 5 | 35 | 7 | 0.875 |
| 454 | 12325 | 8 | 5 | 44 | 8.8 | 1.1 |
| 455 | 12333 | 8 | 5 | 50 | 10 | 1.25 |
| 456 | 12341 | 8 | 5 | 26 | 5.2 | 0.65 |
| 457 | 12349 | 8 | 5 | 20 | 4 | 0.5 |
| 458 | 12357 | 8 | 5 | 24 | 4.8 | 0.6 |
| 459 | 12365 | 8 | 5 | 10 | 2 | 0.25 |
| 460 | 12373 | 8 | 5 | 24 | 4.8 | 0.6 |
| 461 | 12381 | 8 | 5 | 9 | 1.8 | 0.225 |
| 462 | 12389 | 8 | 5 | 5 | 1 | 0.125 |
| 463 | 12397 | 8 | 5 | 13 | 2.6 | 0.325 |
| 464 | 12405 | 8 | 5 | 30 | 6 | 0.75 |
| 465 | 12413 | 8 | 5 | 26 | 5.2 | 0.65 |
| 466 | 12421 | 8 | 5 | 24 | 4.8 | 0.6 |
| 467 | 12429 | 8 | 5 | 29 | 5.8 | 0.725 |
| 468 | 12437 | 8 | 5 | 41 | 8.2 | 1.025 |
| 469 | 12445 | 8 | 5 | 26 | 5.2 | 0.65 |

| 470 | 12453 | 8 | 5 | 27 | 5.4 | 0.675 |
|---|---|---|---|---|---|---|
| 471 | 12461 | 8 | 5 | 31 | 6.2 | 0.775 |
| 472 | 12469 | 8 | 5 | 20 | 4 | 0.5 |
| 473 | 12477 | 8 | 5 | 18 | 3.6 | 0.45 |
| 474 | 12485 | 8 | 5 | 26 | 5.2 | 0.65 |
| 475 | 12493 | 8 | 5 | 38 | 7.6 | 0.95 |
| 476 | 12501 | 8 | 5 | 21 | 4.2 | 0.525 |
| 477 | 12509 | 8 | 5 | 30 | 6 | 0.75 |
| 478 | 12517 | 8 | 5 | 22 | 4.4 | 0.55 |
| 479 | 12525 | 8 | 5 | 17 | 3.4 | 0.425 |
| 480 | 12533 | 8 | 5 | 22 | 4.4 | 0.55 |
| 481 | 12541 | 8 | 5 | 25 | 5 | 0.625 |
| 482 | 12549 | 8 | 5 | 20 | 4 | 0.5 |
| 483 | 12557 | 8 | 5 | 14 | 2.8 | 0.35 |
| 484 | 12565 | 8 | 5 | 13 | 2.6 | 0.325 |
| 485 | 12573 | 8 | 5 | 30 | 6 | 0.75 |

---

## Author Comment (AC2) · 29 Nov 2017

Our previous response to the SC1 had the incorrect links to the NARR datasets that were used in this study. NARR surface variables (e.g. surface precipitation rate and air temperature at the surface) were obtained from NOAA/ESRL Physical Sciences Division (PSD) in Boulder, Colorado (Mesinger et al. 2006; available online at https://www.esrl.noaa.gov/psd/data/gridded/data.narr.monolevel.html). NARR pressure variables (e.g. 500mb geopotential height, 500mb omega, 850mb specific humidity, and vector winds) were also obtained from NOAA/ESRL Physical Sci-

ences Division (PSD) in Boulder, Colorado (Mesinger et al. 2006; available online at https://www.esrl.noaa.gov/psd/data/gridded/data.narr.pressure.html). The pollen data used in this study was obtained by the Neotoma Paleoecology Databse (Carter et al. 2013;2017) available online at http://apps.neotomadb.org/Explorer/?datasetid=22969.

Please note, the charcoal data used in this study (Carter et al. 2013; 2017) will be submitted to the National Climatic Data Center (NCDC) repository hosted by NOAA; https://www.ncdc.noaa.gov/data-access/paleoclimatology-data/contributing.

---

## Referee Comment (RC2) · Anonymous Referee #2 · 21 Dec 2017

General Comments (Overall quality)

This study presents composite anomaly maps derived from the NARR dataset centered on the Upper Platte River Basin for five drought years since 1994. The maps present precipitation rate, temperature, 500 mb geopotential heights and vertical velocity, and 850 mb relative humidity. They indicate anomalously high geopotential heights during the growing seasons of the five drought years, which led to suppressed moisture transport to the region. The climatology methods are sound, results and interpretations are consistent with the data, which support their conclusions regarding the climatic

mechanisms for drought. The paper is generally well written. The results are not unexpected, and as such they are also not particularly novel. However, they do provide sound mechanistic understanding for what the atmosphere does to cause dry weather in this particular region.

The weaknesses of the study largely reside in the sparse explanation of the linkages with a so-called 'mega drought' during the mid-Holocene, for which there is less consensus than the authors convey, both in its spatial homogeneity and temporal expression. Changes are needed for these aspects of the paper, mostly in the form of additional explanation (and citations), more explicit acknowledgment of limitations, adjustments of tone, all to provide the missing information and provide a more informative discussion. Revisions are needed to provide a more accurate description of the paleohydroclimate of the region, and better inform readers about the utility and limitations of modern analogue methods as a diagnostic tool of paleoclimatic data. With these changes, the study will be more accurate and more likely to make a useful contribution.

Specific Comments (individual questions/issues)

1. Incomplete discussion of the regional extent of a so-called '∼150 year long' '4200 Cal BP mega drought' and implications for the utility of seasonal synoptic analogues.

More close attention to precisely what regions this study is meant to be useful for is needed. The Long Lake record, within the Medicine Bow Range, is described here as reflecting the Rocky Mountain region, according to another recently published paper by this author; Carter et al. (2017). However, the citation for the 4.2 ka 'mega' drought is Booth et al. (2005), who focus on the Northern Great Plains. It is not mentioned here that Booth et al.'s hypothesis was not further verified by additional high resolution multi-proxy data (e.g., Grimm et al., 2011). The other records mentioned in support of the drought are Wyoming dune activity and speleothem isotopes from northeastern Utah. However, the dune data is not well-enough dated (OSL and 14C) and conflicting interpretations are possible for the carbon and oxygen speleothem isotopes from

Minnetonka Cave.

Therefore, it is puzzling why the synoptic analyses are focused on the central Rocky mountain region of Wyoming (rather than the Northern Great Plains), and that there is no mention of other paleohydroclimatic data from Wyoming and northern Colorado that are numerous and nearby. Perhaps these regional selections were discussed and justified by Carter et al. (2017) but then this would need to be explained in more detail here. As it is, readers of this study cannot actually evaluate the spatial regional patterns of the modern analogues in relation to any proxy data because it is not shown on the maps. Unfortunately, there are nearby records that do not indicate a 4.2 ka 'mega' drought and which are not mentioned in this study. Through this omission, the study overlooks important implications that likely limit the utility of the modern analogue approach.

2. Incomplete discussion of the temporal uncertainty of drought timing and length and how to understand the relationship between seasonal analogues and lower frequency climate mean states (i.e., multi-decadal to century time-scales).

There is currently no helpful discussion of time-scales in the paper. The range of uncertainty associated with timing of the so called '~150 year' 'mega drought at 4200 Cal BP' is necessary to know in order to contemplate how seasonal anomalies could be translated by radiocarbon dated proxy records. At the very least some discussion of the age control, and uncertainties, for the timing of the quaking aspen rise at Long Lake is needed. The analogues provide seasonal-scale drought mechanisms but discussion about how seasonal synoptic scale mechanisms inform our understanding of drought mechanisms on century time-scales is not here.

3. Incomplete discussion of changing boundary conditions across the 5000 to 4000 Cal BP time window and the potential role of the North American Monsoon (NAM) and El Nino Southern Oscillation (ENSO) that could have potentially affected this study region during that time.

There is no discussion of previous studies based on nearby proxy records that indicate potentially significant changes in the mean state of the NAM and ENSO before and after ∼4 to 3 ka (see Reference list below). Modern day ENSO effects are discounted based on an argument that the region is currently unaffected. The same assumption for the mid-Holocene is likely incorrect. Even if a thorough evaluation of Holocene changes in mean state of NAM and ENSO is beyond the scope of this study, a discussion explaining their potential significance still needs to acknowledged. Changing boundary conditions present major challenges for understanding how to apply modern analogues and should be acknowledged.

4. Sampling of missing relevant references, and references therein: (in no particular order and by no means complete)

-Grimm E.C., Donovan, J.J., Brown, K.J., 2011. A high resolution record of climate variability and landscape response from Kettle Lake, northern Great Plains, North America. QSR 30, 2626-2650.

-Liu, Z. et al. 2014. Paired oxygen isotope records reveal modern North American atmospheric dynamics during the Holocene. Nature Communications 5:3701, doi:10.1038/ncomms4701.

-Higuera, P.E., Briles, C.E., Whitlock, C., 2014. Fire regime complacency and sensitivity to centennial- through millennial-scale climate change in Rocky Mountain subalpine forests, Colorado, USA. Journal of Ecology 102, 1429-1441.

-Anderson, L., Brunelle, A., Thompson, R.S., 2015. A multi-proxy record of hydroclimate, vegetation, fire and post-settlement impacts for a subalpine plateau, central Rocky Mountains, U.S.A. The Holocene 25, 932-943.

-Anderson, L. 2012. Rocky Mountain hydroclimate: Holocene variability and the role of insolation, ENSO, and the North American Monsoon. Global and Planetary Change 92-93, 198-208.

-Whitlock et al., 2012. Holocene seasonal variability inferred from multiple proxy records from Crevice Lake, Yellowstone National Park, USA. P3 331-332, 90-103

-Shuman, B.N., Marsicek, J., 2016. The structure of Holocene climate change in mid-latitude North America. QSR 141, 38-51.

Technical Corrections (typing errors, grammar etc.)

-As previous reviewer suggested, avoid emotive language and delete "Unfortunately" on lines 5 and 13. -p.5 Line 24, spelling of "analyse" -p.9 Line 24, "of flow of cold"?

---

## Author Comment (AC3) · 17 Jan 2018

Dear Reviewer #1. Thank you for your input and suggestions. We believe your suggestions have improved the manuscript.

1. The composites are based on only five events. But there is no attempt anywhere in the paper to address the statistical significance or the robustness of the results. The features of the maps may easily in many cases be just results of chance. This should be investigated by calculating and showing the statistical significance. Also, it should

be tested if the results are robust and if they depend on one or a few of the five events. It should also be tested if results depend on the threshold (-1.5 standard deviations).

Response to comment #1: To justify our selection of analogue years, we used a two-tailed Student's t-test with an alpha of 0.05 to quantify significance at each grid point in the data set comparing the precipitation that fell during our five composite years to that during the 30-year climate normal (1981-2010). Because climate composites are calculated using an arithmetic mean (30-year climate normal) as the measure of central tendency, we feel a two-tailed Student's t-test was the best statistical test to validate the significance of our case years. Our results are then presented in a map depicting the spatial distribution of significant p-values ($p < 0.05$) across our study region. We propose to include this new map as Figure 2b (see proposed Fig. 2b below). Precedent for using a t-test to calculate statistical significance of composite anomalies in climatological analyses is well established in the existing literature (see Cayan 1996; Shabbar and Khandekar 1996; Taschetto and England 2009). Spatially, if we look at the most statistically significant case years ($p < 0.05$) from the modern record (e.g. the shared common period between the precipitation data and NARR data between 1979 – present), the significance values are representative of persistent conditions which we use as modern climate analogues in our analyses.

2. The duration of the modern analogues are around a year, while the duration of the mega drought is more than 100 years. Is there any reason at all to believe that events on such different time-scales have the same or related mechanisms? Long lasting events tend, in general, to also be more spatially extended. See e.g. DOI: 10.1002/2016RG000521 for a review of how the number of spatial degrees of freedom depends on the temporal scale considered. The validity of the method of modern analogues should be investigated and discussed in detail.

There is a lot of available model experiments (e.g. CMIP5) where this could be investigated.

[Figure]

Response to comment #2: While the duration of the individual case years used in our analyses represent one year, the composite-anomaly approach takes into consideration the overall processes occurring within the selected group of case years as representative of persistent conditions, providing an analogue to the long-term drought identified in the paleoecological record. Your statement 'Long lasting events tend, in general, to also be more spatially extended' is illustrated in the results of the Student's t-test described above, demonstrating that that our case years were not only anomalously dry in our study region of south-eastern Wyoming, but our case years were anomalously (and statistically significant) dry across the region (see proposed Fig. 2b below). Thank you for suggesting the Christiansen and Ljungqvist (2017) reference. While we agree the degrees of freedom should be considered in temperature reconstructions, our paper is not attempting to reconstruct temperature – rather the goal of the modern climate analogue approach is to understand atmospheric processes involved with prolonged drought that then potentially can be used to explain ecological responses to persistent drought conditions. We propose adding a paragraph to the Introduction which provides additional literature discussing previous studies that have used the modern climate analogue technique. We feel that the addition of this new paragraph will address the reviewer's comments, and provide the reader with the proper background and citations that justifies the use of modern climate analogues as a potential way to explain past atmospheric processes and conditions. Our analysis is a first-step approach to understanding the relationship between vegetation change and drought-related disturbance at a local and regional scale. Since the NARR data are initiated with climate station data, our study can be used for data-model comparisons utilizing climate models (e.g. CMIP5), as well as sensitivity tests of the described processes in future analyses.

Minor comments: p6, l13: Why are these years "suitable analogues". Are other conditions than the drought index used?

Response for comment p6, l13: We apologize if this text was vague. What we meant

by 'suitable analogues' is that the five case years identified represent the driest years (-1 standard deviation or below from the mean) in the modern record. However, using the results from the Student's t-test, we are able to say that our case years are statically robust analogues, rather than just stating they are 'suitable.' We will update the sentence on p6, l13 to say, 'Five case years (2012, 2002, 2001, 1988, and 1994) that were -1 standard deviations below the long-term average were chosen because they were found to be the most statistically significant ($p < 0.05$) analogues for dry conditions in the North Platte River Basin.'

Figure 2: What is the value of the standard deviation?

Response for Figure 2: One standard deviation is equivalent to 58.89 mm. We would like to point out that the Y axis on Figure 2 was previously mislabeled. The Y axis on the figure presented in the attachment is now correct. We also would like to point out another error that was in the manuscript. We previously described analogues as being suitable if they were -1.5 standard deviations from the mean. However, the correct number should be -1 standard deviations from the mean. We apologize for these errors.

Additional References related to our analysis of statistical significance:

Cayan, D. R. (1996). Interannual climate variability and snowpack in the western United States. Journal of Climate, 9(5), 928-948.

Shabbar, A., & Khandekar, M. (1996). The impact of el Nino‐Southern oscillation on the temperature field over Canada: Research note. Atmosphere-Ocean, 34(2), 401-416.

Taschetto, A. S., & England, M. H. (2009). El Niño Modoki impacts on Australian rainfall. Journal of Climate, 22(11), 3167-3174.

[Figure]

Figure 2. Precipitation anomalies and the spatial distribution of significant p-values across the study region of south-eastern Wyoming. A) A time series of annual precipitation anomalies for 1979-2014 compared to the long-term average (1981-2010) from Wyoming climate division 10, Upper Platte River Basin. The first five years with -1 or more standard deviations below the long-term average include 2012, 2002, 2001, 1998, and 1994. One standard deviation equates to 58.89 mm. Climate division data were collected from http://www.esrl.noaa.gov/psd/cgi-bin/timeseries/timeseries1.pl. B) A map showing the spatial distribution of significant p-values (p < 0.05) across the study region (outlined in red box) identified during the five driest years. P-values were evaluated using a two-tailed Student's t-test with an alpha of 0.05.

**Fig. 1.**

---

## Author Comment (AC4) · 17 Jan 2018

Dear Reviewer #2. Thank you for your input and suggestions. We believe your suggestions will greatly improve the manuscript.

1. Incomplete discussion of the regional extent of a so-called '150 year long' '4200 Cal BP mega drought' and implications for the utility of seasonal synoptic analogues. More close attention to precisely what regions this study is meant to be useful for is needed. The Long Lake record, within the Medicine Bow Range, is described here as

reflecting the Rocky Mountain region, according to another recently published paper by this author; Carter et al. (2017). However, the citation for the 4.2 ka 'mega' drought is Booth et al. (2005), who focus on the Northern Great Plains. It is not mentioned here that Booth et al.'s hypothesis was not further verified by additional high resolution multi-proxy data (e.g., Grimm et al., 2011). The other records mentioned in support of the drought are Wyoming dune activity and speleothem isotopes from northeastern Utah. However, the dune data is not well-enough dated (OSL and 14C) and conflicting interpretations are possible for the carbon and oxygen speleothem isotopes from Minnetonka Cave.

Therefore, it is puzzling why the synoptic analyses are focused on the central Rocky mountain region of Wyoming (rather than the Northern Great Plains), and that there is no mention of other paleohydroclimatic data from Wyoming and northern Colorado that are numerous and nearby. Perhaps these regional selections were discussed and justified by Carter et al. (2017) but then this would need to be explained in more detail here. As it is, readers of this study cannot actually evaluate the spatial regional patterns of the modern analogues in relation to any proxy data because it is not shown on the maps. Unfortunately, there are nearby records that do not indicate a 4.2 ka 'mega' drought and which are not mentioned in this study. Through this omission, the study overlooks important implications that likely limit the utility of the modern analogue approach.

Response to comment #1: Thank you for your suggestion. We agree there are a variety of published records within the region (central Rocky Mountains and northern Great Plains) that document conflicting accounts of paleohydroclimate between 5,000 and 3,500 cal yr BP. However, the published paleo-proxy climate reconstructions rarely, if at all, provide insight into synoptic processes and mechanistic conceptual models such as those we are presenting in our study. We propose adding a section in the discussion that addresses paleohydroclimate in the central Rocky Mountains and northern Great Plains region in the context of modern climate analogues.

2. Incomplete discussion of the temporal uncertainty of drought timing and length and how to understand the relationship between seasonal analogues and lower frequency climate mean states (i.e., multi-decadal to century time-scales). There is currently no helpful discussion of time-scales in the paper. The range of uncertainty associated with timing of the so called '_150 year' 'mega drought at 4200 Cal BP' is necessary to know in order to contemplate how seasonal anomalies could be translated by radiocarbon dated proxy records. At the very least some discussion of the age control, and uncertainties, for the timing of the quaking aspen rise at Long Lake is needed. The analogues provide seasonal-scale drought mechanisms but discussion about how seasonal synoptic scale mechanisms inform our understanding of drought mechanisms on century time-scales is not here.

Response to comment #2: We propose to include a sentence in the Methods section that clearly answers the point regarding the timing of the quaking aspen period at Long Lake identified by Carter et al. (2017). As for the discussion pertaining to how seasonal synoptic scale mechanisms inform our understanding of drought mechanisms on century time-scales, we point out that our aim is not to reconstruct climate during the drought period identified by Carter et al. (2017). Rather, the point of our paper is to investigate plausible mechanisms that could be used to explain the ecological changes from the central Rocky Mountains. Similar to Shinker et al. (2006), we selected modern climate analogues to investigate whether the analogues can be used to infer processes that were likely important in controlling moisture anomalies in the past. Using the principle of uniformitarianism, we believe that the mechanisms that cause droughts at present (on seasonal, annual, and decadal time-scales) can be used to explain paleo droughts. Further, our composite-anomaly analyses use statistically significant dry years which represent persistent dry conditions as analogues for climate mechanisms and processes associated with drought conditions in the past.

3. Incomplete discussion of changing boundary conditions across the 5000 to 4000 Cal BP time window and the potential role of the North American Monsoon (NAM) and El

Nino Southern Oscillation (ENSO) that could have potentially affected this study region during that time.

There is no discussion of previous studies based on nearby proxy records that indicate potentially significant changes in the mean state of the NAM and ENSO before and after _4 to 3 ka (see Reference list below). Modern day ENSO effects are discounted based on an argument that the region is currently unaffected. The same assumption for the mid-Holocene is likely incorrect. Even if a thorough evaluation of Holocene changes in mean state of NAM and ENSO is beyond the scope of this study, a discussion explaining their potential significance still needs to acknowledged. Changing boundary conditions present major challenges for understanding how to apply modern analogues and should be acknowledged.

Response to comment #3: Thank you for your suggestion. We propose to add additional information based on the suggested references, as well as additional ones in the discussion section with regards to changing boundary conditions across 5,000 and 4,000 cal yr BP. We will discuss the role of the North American Monsoon and ENSO on the region at that time to strengthen this work.

4. Sampling of missing relevant references, and references therein: (in no particular order and by no means complete)

Response to comment #4: Thank you for the suggested references. We will include the suggested references, and references therein in our newly proposed discussion sections.

Technical Corrections (typing errors, grammar etc.) -As previous reviewer suggested, avoid emotive language and delete "Unfortunately" on lines 5 and 13. -p.5 Line 24, spelling of "analyse" -p.9 Line 24, "of flow of cold"?

Response to technical corrections: Thank you for pointing out these typing errors. We will avoid emotive language, delete "Unfortunately" on lines 5 and 13, and we will delete

'of' on p.9 l24.

---

## Author Response (AR1)

**Drought and vegetation change in the central Rocky Mountains: Potential climatic mechanisms associated with  drought conditions at 4200 cal yr BP.**

Vachel A. Carter[1,2], Jacqueline J. Shinker[3], Jonathon Preece[3]

[1]RED Lab, Department of Geography, University of Utah, Salt Lake City, UT, 84112, USA
[2]Department of Botany, Charles University, Prague, 12801, Czech Republic
[3]Department of Geography and Roy J. Shlemon Center for Quaternary Studies, University of Wyoming, Laramie, WY, 82071, USA

*Correspondence to*: Vachel A. Carter (vachel.carter@gmail.com)

We would like to thank both reviewers and the Editor for their valuable comments. Below you will find our response to each reviewer, as well as the Editor, and highlight all changes that have been made to the manuscript. Our responses are shown in blue text.

**Reviewer 1**

The composites are based on only five events. But there is no attempt anywhere in the paper to address the statistical significance or the robustness of the results. The features of the maps may easily in many cases be just results of chance. This should be investigated by calculating and showing the statistical significance. Also, it should be tested if the results are robust and if they depend on one or a few of the five events. It should also be tested if results depend on the threshold (-1.5 standard deviations).

We thank Reviewer 1 for pointing out this specific weakness in the paper. We now include which statistical significance test we used, and justify the use of this particular significance test in the Methods section (please see page 16 lines 7-12 below). Specifically, we used a two-tailed Student's t-test with an alpha of 0.05 to quantify significance at each grid point in the data set by comparing the precipitation that fell during our five composite years to that during the 30-year climate normal (1981-2010). We have demonstrated that spatially, if we look at the most statistically significant case years ($p < 0.05$) (i.e. those that were -1 standard deviations from the mean) from the modern record (e.g. the shared common period between the precipitation data and NARR data between 1979 and present), the significance values are representative of persistent drought conditions which we use as modern climate analogues in our analyses. To illustrate these results, we also included an additional figure (Figure 2b; see below) and associated text. This additional figure depicts the spatial distribution of significant p-values ($p < 0.05$) (i.e. those at the -1 standard deviation threshold) across our study region. The plotted significance values in Figure 2b indicate spatial coherency of anomalous dry conditions rather than spurious individual grid cells that would likely occur from

chance, further supporting the use of our selected years as representative of both persistence (in time) and consistence (in space) of anomalous dry conditions.

To test whether the composite-anomaly values were dependent upon one of the constituent within the five events, we reviewed individual seasonal composite values within each of the five case years, and with the exception of DJF in 2012 and 1988, all

5  of the anomalous precipitation values are consistently below normal with regards to precipitation, indicating that our composites dominated by representative persistent anomalously dry conditions. Thus, our composite values are not dependent upon one or a few of the seasons within the five event years. Additionally, it has been demonstrated in the literature (Mock and Brunelle-Daines, 1999; Mock and Shinker, 2013) that the use of a single analogue case year or even month is robust to discuss synoptic processes in the past.

[Figure]

Figure 2. Precipitation anomalies and the spatial distribution of significant p-values across the study region of south-eastern Wyoming. A) A time series of annual precipitation anomalies for 1979-2014 compared to the long-term average (1981-2010) from Wyoming climate division 10, Upper Platte River Basin. The first five years with -1 or more standard deviations below the long-term average include 2012, 2002, 2001, 1998, and 1994. One standard deviation equates to 58.89 mm. Climate division data were collected from http://www.esrl.noaa.gov/psd/cgi-bin/timeseries/timeseries1.pl. B) A map showing the spatial distribution of significant p-values (p < 0.05) across the study region (outlined in red box) identified during the five driest years. P-values were evaluated using a two-tailed Student's t-test with an alpha of 0.05.

The duration of the modern analogues are around a year, while the duration of the mega drought is more than 100 years. Is there any reason at all to believe that events on such different time-scales have the same or related mechanisms? Long lasting

15  events tend, in general, to also be more spatially extended. See e.g. DOI:10.1002/2016RG000521 for a review of how the

number of spatial degrees of freedom depends on the temporal scale considered. The validity of the method of modern analogues should be investigated and discussed in detail. There is a lot of available model experiments (e.g., CMIP5) where this could be investigated.

The underlying assumption in the modern climate analogue approach is in the principle of uniformitarianism (Barry and Perry, 1973), which assumes that the range of modern conditions contains extreme events that likely occurred in the past, and that the processes involved with modern synoptic processes behaved similarly in the past. It has been demonstrated that multi-decadal to centennial-scale droughts were common phenomena in the Great Plains region during the late Holocene (Schmieder et al., 2011). Therefore, we believe that our composite-anomaly approach takes into consideration the overall processes occurring within the five case years, when combined, as representative of persistent conditions, providing an analogue to the long-term drought identified in the paleoecological record. We discuss the principle of uniformitarianism in the Introduction (page 13 lines 8 below) and discussion section 5.3 (page 26 lines 2 below).

Your statement 'Long lasting events tend, in general, to also be more spatially extended' is illustrated in the additional figure (Figure 2b) which demonstrates that that our case years were not only anomalously dry in our study region of south-eastern Wyoming, but our case years were anomalously (and statistically significant) dry across the region. Thank you for suggesting the Christiansen and Ljungqvist (2017) reference. While we agree the degrees of freedom should be considered in temperature reconstructions, our paper is not attempting to reconstruct temperature, which does have a normal distribution because of the influence of seasonal insolation – rather the goal of the modern climate analogue approach is to understand atmospheric processes involved with prolonged drought associated with precipitation, which is not normally distributed because of the variability of uplift mechanisms and moisture availability, which we use in our approach to analyze and explain the climatic controls of ecological responses to persistent drought conditions.

We have included a new paragraph in the Introduction section that discusses the validity of the modern analogue method. In this new paragraph, we provide a brief literature review on previous research that used the modern climate analogue technique as a way to explain past atmospheric processes associated with a variety of environmental change. This new paragraph can be found on page 13 line 20.

Our analysis is a first-step approach to understanding the relationship between vegetation change and drought-related disturbance at a local and regional scale. Since the NARR data are initiated with climate station data, our study can be used for data-model comparisons utilizing climate models (e.g. CMIP5), as well as sensitivity tests of the described processes in future analyses. While we did not include model simulations in this study, we have included a new discussion section, 5.3, titled 'Model simulations and implications for drought in the central Rocky Mountains and central Great Plains' where we briefly discuss previous research that have used GCMs and RegCMs to investigate and model drivers of drought in the region. Our results agree with existing model simulations that anomalous high-pressure ridges and anti-cyclonic flow are prominent

features involved with drought in the region at multiple timescales (e.g. mid-Holocene, historical Dust Bowl, recent modern drought within the timeframe of our study). Section 5.3 begins on page 25 line 12.

***Reviewer 1 Minor comments***

p6, 113: Why are these years "suitable analogues". Are other conditions than the drought index used?

We apologize if this text was vague. What we meant by 'suitable analogues' is that the five case years identified represent the driest years (i.e. -1 standard deviations below the mean) in the modern record. However, using the results from the Student's t-test, we are able to say that our case years are statically robust analogues, rather than just stating they are 'suitable.' The sentence has been edited to read: "Five case years (2012, 2002, 2001, 1988, and 1994) that were at least -1 standard deviations below the long-term average were chosen because they were found to be the most statistically significant ($p < 0.05$) analogues for dry conditions in the Upper Platte River Basin (Figure 2b). These five case years are representative of persistently low annual precipitation conditions and are therefore used to as analogues to explain the climatic mechanisms responsible for dry conditions identified in the paleoclimate record." This can be found on page 17 line 5.

Figure 2: What is the value of the standard deviation?

We apologize we did not include this information. One standard deviation is equivalent to 58.89 mm per year. This information has been included in both the Methods section (page 16 line 7 below), and in the new Figure 2b figure caption (see above).

**Reviewer 2**

Incomplete discussion of the regional extent of a so-called '_150 year long' '4200 Cal BP mega drought' and implications for the utility of seasonal synoptic analogues. More close attention to precisely what regions this study is meant to be useful for is needed. The Long Lake record, within the Medicine Bow Range, is described here as reflecting the Rocky Mountain region, according to another recently published paper by this author; Carter et al. (2017). However, the citation for the 4.2 ka 'mega' drought is Booth et al. (2005), who focus on the Northern Great Plains. It is not mentioned here that Booth et al.'s hypothesis was not further verified by additional high resolution multi-proxy data (e.g., Grimm et al., 2011). The other records mentioned in support of the drought are Wyoming dune activity and speleothem isotopes from northeastern Utah. However, the dune data is not well-enough dated (OSL and 14C) and conflicting interpretations are possible for the carbon and oxygen speleothem isotopes from Minnetonka Cave.

Therefore, it is puzzling why the synoptic analyses are focused on the central Rocky mountain region of Wyoming (rather than the Northern Great Plains), and that there is no mention of other paleohydroclimatic data from Wyoming and northern Colorado that are numerous and nearby. Perhaps these regional selections were discussed and justified by Carter et al. (2017) but then this would need to be explained in more detail here. As it is, readers of this study cannot actually evaluate the spatial regional patterns of the modern analogues in relation to any proxy data because it is not shown on the maps. Unfortunately, there are

nearby records that do not indicate a 4.2 ka 'mega' drought and which are not mentioned in this study. Through this omission, the study overlooks important implications that likely limit the utility of the modern analogue approach.

Thank you for your suggestion. We have expanded our discussion to reflect the regional complexities of climate variability both spatially and temporally. Due to the proximity of our study region in south-eastern Wyoming to the central Great Plains, our composite anomaly maps illustrate that the broad-scale synoptic processes that impact the central Great Plains also influence our study region. Therefore, we focus on proxy evidence closest to our study region. Because Booth et al. (2005) included sites in both the central and northern Great Plains, and the additional text we have included supports the climatic connections between the Great Plains region and the eastern portion of the central Rocky Mountains.

We now include a new discussion section, section 5.1 'Proxy evidence for regional climate variability at 4200 cal yr BP' (please see page 19 line 10 below) which includes additional sites in the central Rocky Mountains and central Great Plains region that also experienced ecological change in response to drought conditions around 4200 cal yr BP. We also discuss sites near our study region that do not record drought conditions at this time. The conflicting climate interpretations from the region (i.e. cool and wet versus warm and dry) is to be expected because our single site cannot reconstruct similar conditions found in the northern Great Plains and interior intermountain west which have different precipitation regimes and synoptic controls, and are topographically more complex and lacking proximity to Great Plains climatic influences. Additionally, contrasting evidence in the region can be explained by differences in site and proxy sensitivities, seasonal biases in proxy, as well as temporal resolution of each individual site.

Incomplete discussion of the temporal uncertainty of drought timing and length and how to understand the relationship between seasonal analogues and lower frequency climate mean states (i.e., multi-decadal to century time-scales). There is currently no helpful discussion of time-scales in the paper. The range of uncertainty associated with timing of the so called '_150 year' 'mega drought at 4200 Cal BP' is necessary to know in order to contemplate how seasonal anomalies could be translated by radiocarbon dated proxy records. At the very least some discussion of the age control, and uncertainties, for the timing of the quaking aspen rise at Long Lake is needed. The analogues provide seasonal-scale drought mechanisms but discussion about how seasonal synoptic scale mechanisms inform our understanding of drought mechanisms on century time-scales is not here.

Thank you for pointing out the discrepancy in temporal uncertainty. We discuss the length and timing of the drought in the Introduction section on page 12 line 26. We expand upon the temporal certainty regarding the timing of the drought at Long Lake in the Study Area section on page 15 line 14 below.

The underlying assumption of using the modern climate analogue approach is in the principle of uniformitarianism (Barry and Perry, 1973), which assumes that the range of modern conditions contains extreme events that may have been more frequent in the past, and that the processes involved with modern synoptic controls behaved similarly in the past. Our composite-anomaly analyses use statistically significant dry years which represent persistent (in time) dry conditions as analogues for climate mechanisms and processes associated with drought conditions in the past. While Carter et al. (2017a) were unable to

confirm whether the drought between 4300 and 4100 cal yr BP was indeed ~150-years long because of low temporal resolution involved with slow sediment accumulation rates, it is more plausible that this drought occurred on multi-decadal time-scales. However, both multi-decadal- and centennial-scale droughts were common phenomena in the Great Plains and western US during the late Holocene (Woodhouse and Overpeck, 1998; Cook et al., 2004; Schmieder et al., 2011; Cook et al., 2016).

5    Regardless, the modern climate analogues are representative of persistent drought conditions and thus could be reflective of droughts on both multi-decadal and centennial- time-scales given the consistency in synoptic controls (persistent anomalous high pressure supporting subsidence, and block moisture flow into the region). Our modern climate analogue results include synoptic patterns that are consistent with other reconstructed droughts in the Great Plains region during a variety of times (e.g. mid-Holocene, historical Dust Bowl, and recent modern droughts).

10    Incomplete discussion of changing boundary conditions across the 5000 to 4000 Cal BP time window and the potential role of the North American Monsoon (NAM) and El Nino Southern Oscillation (ENSO) that could have potentially affected this study region during that time.

There is no discussion of previous studies based on nearby proxy records that indicate potentially significant changes in the mean state of the NAM and ENSO before and after _4 to 3 ka (see Reference list below). Modern day ENSO effects are

15    discounted based on an argument that the region is currently unaffected. The same assumption for the mid-Holocene is likely incorrect. Even if a thorough evaluation of Holocene changes in mean state of NAM and ENSO is beyond the scope of this study, a discussion explaining their potential significance still needs to acknowledged. Changing boundary conditions present major challenges for understanding how to apply modern analogues and should be acknowledged.

Thank you for your suggestion. We have rewritten section 5.1 which as previously mentioned, discusses the regional climate

20    variability between 5000 and 4000 cal yr BP as evidenced from regional proxy data. Also included in this new discussion section are discussions regarding changing boundary conditions that may have caused the different spatial patterns in climate. The new 5.1 discussion section begins on page 19 line 10 below. We provide additional context and additional references for the role, or lack thereof, of changes and impacts of ENSO and the NAM on our region, and the relationship to synoptic patterns associated with our modern climate analogue processes.

Sampling of missing relevant references, and references therein: (in no particular order and by no means complete)

Thank you for the suggested references. We have included the suggested references, and references therein in our new discussion sections.

***Reviewer 2 Minor comments***

30    Technical Corrections (typing errors, grammar etc.) -As previous reviewer suggested, avoid emotive language and delete "Unfortunately" on lines 5 and 13. -p.5 Line 24, spelling of "analyse" -p.9 Line 24, "of flow of cold"?

Thank you for your suggestions. We deleted 'unfortunately' from page 12 line 1 and line 12, as well as on page 29 line 12. We corrected the spelling of 'analyse' on page 16 line 27 below.

**Editor's comments**

Dear Carter et al, Thank you very much for submitting your manuscript to the YSM special issue of Climate of the Past and for your responses to the reviewer's comments. We would like to submit your paper for major revisions and a second round of review. We agree with the reviewer's comments and look forward to your planned edits of the manuscript in line with their suggestions.

We would like to thank the Handling Editor and Co-Editor for the opportunity to revise and resubmit our manuscript, cp-2017-107: "Drought and vegetation change in the central Rocky Mountains: Potential climatic mechanisms associated with mega drought conditions at 4200 cal yr BP." We are appreciative of the comments from the two anonymous reviewers. Their comments have greatly improved the discussion and applicability of the modern climate analogue technique in regards to the drought at 4200 cal yr BP.

We have slightly changed the title of our manuscript in this submission: ' Drought and vegetation change in the central Rocky Mountains: Potential climatic mechanisms associated with  drought conditions at 4200 cal yr BP' because of Reviewer 2 comments. Carter et al. (2017a) were able to reconstruct a drought between 4300 and 4100 cal yr BP, but because of temporal resolution reasons, the authors were unable to determine whether the drought was indeed a 'megadrought.' We therefore have removed 'mega' from our title.

We have included a new co-author, Jonathon Preece. His affiliation has been included in the revised manuscript.

Per the major revision recommendation, we have completely rewritten the discussion section; section 5.1 'Proxy evidence for regional climate variability at 4200 cal yr BP' addresses Reviewer 2 Comment 1 and 3; section 5.2 'Regional climate variability based on modern climate analogues of drought at 4200 cal yr BP' uses the modern climate analogue approach to explain the ecological response in our study region of south-eastern Wyoming, as well as within the central Great Plains; finally, section 5.3 'Model simulations and implications for drought in the central Rocky Mountains and central Great Plains' addresses Reviewer 1 Comment 2 regarding the use of model simulations. While we did not include simulations in this paper (this will be included in a follow-up paper), we did compare the results of our modern climate analogue approach to existing literature that have used GCMs and RegCMs to investigate drivers of drought in the region. Our results are supportive of existing models and simulations that suggest persistent high-pressure ridges and anti-cyclonic winds are prominent features in droughts from the region.

As additional editorial advice, we also request that in response to Reviewer 1 Comment 1 the method of testing for statistical significance should be included in the Data and Methods section of the paper. This section should include a response to the question of whether your results are sensitive to exclusion of any of the five composite members and to the choice of -1 standard deviation threshold. To address Reviewer 1 Comment 2, the addition of an introductory paragraph on the climate

5   analogue approach is welcomed.

We used a Student's t test to address Reviewer 1 Comment 1, which we now discuss in the Methods section on page 16 line 7 below. Using this statistical analysis, we were able to confirm that -1 standard deviations is a statistically significant, representing persistent drought conditions through time. We plotted the resulting significance values to illustrate their spatial coherence to illustrate the valid threshold used in our study. This can be found in the Methods section on page 16.

The netcdf files used for the statistical analyses, as well as for creating the composite anomaly maps, will be uploaded to PANGAEA.

Additionally, it has been demonstrated in the literature (Mock and Brunelle-Daines, 1999; Mock and Shinker, 2013) that the

15  use of a single analogue case year is robust to discuss synoptic processes in the past. We did investigate whether our composite-anomaly values were dependent upon one of a few of the seasons within the five case years selected by reviewing individual seasonal composite values within each of the five selected case years. With the exception of DJF in 2012 and 1988, all of the anomalous precipitation values are consistently below normal with regards to precipitation, indicating that our composites are representative of persistent anomalously dry conditions (also evident in the significance values form the time series). We

20  included this information in the discussion section on page 22 lines 20.

Lastly, we have also included an introductory paragraph regarding the climate analogue approach. This new paragraph can be found on page 13.

25  We further request that the assumptions underlying the climate analogue approach should also be discussed in the Discussion section and inform the Conclusions.

We now discuss the underlying assumption involved with the modern climate analogue technique in the new 5.3 discussion section (see page 25 line 12). The underlying assumption was used to inform the Conclusion section which begins on page 27.

30  We would also request that you address the suggestion to use CMIP5 simulations to test the assumptions in the MAT.

We believe climate simulations like CMIP5 and other GCMs and RCMs are beneficial for understanding climate variability and teleconnection impacts in areas sensitive to drought. Currently, there are several leading hypotheses regarding the mechanism behind the 4.2 ka event, but no climate simulations exist for this time period. The goal of using an environment-to-circulation approach using the modern climate analogue technique was to investigate synoptic processes that may be used

to explain the ecological changes we see in south-eastern Wyoming at 4200 cal yr BP. The results of this study are meant to inform future climate simulations regarding this climatic event, not reconstruct and model climate processes involved with initial cause of the event. Future work will involve data-model comparisons to test whether the results presented, while consistent with past regional drought reconstructions (e.g. mid-Holocene, historical Dust Bowl, and modern droughts) are

5    useful for reconstructing boundary conditions to improve our understanding of the 4.2ka event.

To help address Reviewer 2 Comment 2, we suggest removing Section 3.1 from the paper and expanding on the length and timing of the dry spell in the Introduction instead.

Thank you for the suggestion. We have removed section 3.1, but have used some of the relevant information in the end of the

10  study area section (see page 15). There, we expanded upon Reviewer 2 Comment 2 regarding the temporal constraint of the drought identified in our sedimentary record. Additionally, we briefly expand on the length and timing of the drought in the Introduction section (see page 12 line 26).

For Reviewer 2 Comment 3, we advise using the suggested references to develop your discussion further in light of other

15  studies conducted in the area, and particularly in the context of your revised Figure 2b.

We thank Reviewer 2 for the suggested references and used these papers in our new discussion section, 5.1 which begins on page 19. We have provided context for the differences in proxy and site sensitivities within the region, which is to be expected given the topographic diversity of the interior intermountain west, as well as variability with synoptic processes (e.g. variations in the influence of the polar jet stream). In other words, topographically complex regions such as the Rocky Mountains and

20  intermountain west region experiences different precipitation regimes (seasonally and spatially) that can impact the ecological signals recorded in proxy data at a variety of sites. We discuss this further in section 5.1.

**References cited in author response**

[revised manuscript text omitted]

---

## Referee Report (RR1)

Comment 1:
This paper is mainly introducing a statistical method to explore climate mechanisms, which is a good idea. Generally speaking, this paper is well organized with 1) a presentation of the model simulations, 2) a discussion and comparison of the regional proxies, 3) a summary of the model implications.

Comment 2:
The conclusion is rather long. The middle paragraph should be removed from there, and possibly merged in the discussion.

Comment 3:
Line 26: "Yet, the exact timing of drought conditions based on different proxy varies spatially and temporally" needs more details. For example, what are the main causes for the spatial and temporal variations?

Comment 4:
P9-line 5: Need a brief introduction to "Bond event 3".

Additional comments.
1. You use the modern 5-anomaly dry years as representation to discuss the 4200 years climate mechanism. Does it mean that you can draw the same conclusion for the other drought events in your sedimentary core if there is any?
2. How do you deal with the fact that while around 4200 the vegetation is dominated by Populus, and that now it is dominated by conifers. That means the climatic conditions are very different, yet your reconstruction of the 4.2 dry event is based on pollen.

---

## Editor Decision (ED1)

Dear Carter et al,

Thank you very much for submitting your manuscript to the YSM special issue of Climate of the Past and for your responses to the reviewer's comments. We would like to submit your paper for major revisions and a second round of review. We agree with the reviewer's comments and look forward to your planned edits of the manuscript in line with their suggestions.

As additional editorial advice, we also request that in response to Reviewer 1 Comment 1 the method of testing for statistical significance should be included in the Data and Methods section of the paper. This section should include a response to the question of whether your results are sensitive to exclusion of any of the five composite members and to the choice of -1 standard deviation threshold.

To address Reviewer 1 Comment 2, the addition of an introductory paragraph on the climate analogue approach is welcomed. We further request that the assumptions underlying the climate analogue approach should also be discussed in the Discussion section and inform the Conclusions. We would also request that you address the suggestion to use CMIP5 simulations to test the assumptions in the MAT.

To help address Reviewer 2 Comment 2, we suggest removing Section 3.1 from the paper and expanding on the length and timing of the dry spell in the Introduction instead.

For Reviewer 2 Comment 3, we advise using the suggested references to develop your discussion further in light of other studies conducted in the area, and particularly in the context of your revised Figure 2b.

We would especially like to thank the reviewer's for their insightful and substantive comments which will ultimately make this a stronger paper. We look forward to seeing the revised manuscript.

Best wishes,
Heather (Handling Editor) and Mike Evans (Co-Editor)

---

## Author Response (AR2)

**Response to Editor**

We would like to sincerely thank you for allowing us to resubmit our manuscript. We feel that this version has addressed all the major problems pointed out by Referee's #3, #4, and #5. Specifically, in this version, we have expanded our surface paleo response to include the western Great Plains region as
several major sand dunes reactived around the same time we see unique changes in vegetation composition at our study site. As such, we redid our analyses in order to find the best analogues that best represent the paleo-surface conditions that occurred ~4200 cal yr BP. Lastly, as requested, we now have a new discussion section, '4.3 Limitation of the modern climate analogue technique' which specifically addresses the limitations of using the modern climate analogue technique as a method for
analyzing past synoptic dynamics. As pointed out by Referee #4, the largest assumption, in our case, is that our two analogues are reflective of megadrought conditions that occur on decadal-to-centennial timescales. In addition, we now express more uncertainty in the language of our manuscript. We hope it is now clearer in the language that this method simply offers a scenario of the 4.2 ka drought in our study region. We hope that our research presents an opportunity for the modeling community to test
whether the results of our study are valid representations of megadrought conditions on decadal-to-centennial timescales.

**Response to Reviewer #3**

Dear Reviewer #3. Thank you for your kind comments in Comment #1, as well as your input and
suggestions in Comments #2 through 4. We believe your suggestions have improved the manuscript.

Comment #2: The conclusion is rather long. The middle paragraph should be removed from there, and possibly merged in the discussion section.

Response to comment #2: Thank you for your suggestion. We have deleted the middle paragraph from
the conclusion section, and incorporated it into the discussion section where needed.

Comment #3: Line 26 "Yet, the exact timing of drought conditions based on different proxy varies spatially and temporally" needs more details. For example, what are the main causes for the spatial and temporal variations?

Response to comment #3: We apologize this was unclear. The previous paragraph was supposed
articulate how the different proxies discussed varied spatially and temporally. We have removed this sentence from the manuscript since it was confusing for the reader.

Comment #4: P9-line 5: Need a brief introduction to 'Bond event 3".

Response to comment #4: Per recommendation to stream line the discussion, we have deleted segments of the discussion, including the paragraph this sentence was included in. Thus, we did not introduce Bond events in the manuscript.

Additional comment #1: You use the modern 5-anomaly dry years as representation to discuss the 4200 years climate mechanism. Does it mean that you can draw the same conclusion for the other droughts in your sedimentary core if there is any?

Response to Additional comment #1: Yes, we can draw the same conclusion. Carter et al. (2017a) also observed this same lagged vegetation response in response to the 1930s or 1950s drought. The 1930s and 1950s droughts experienced anomalous high-pressure ridges over the central US. Anomalous high-pressure ridges were experienced in our analogues. Thus, we hypothesize that the synoptic processes present during the 1930s/1950s drought must have been present during the 4.2ka drought as both cases resulted the brief increase in quaking aspen.

Additional comment #2: How do you deal with the fact that while around 4200 the vegetation is dominated by Populus, and that now it is dominated by conifers. That means the climatic conditions are very different, yet your reconstruction of the 4.2 dry event is based on pollen.

Response to Additional comment #2: We believe the previous response also helps answer this comment. You are correct in that the vegetation composition is currently different, yet, the aspen ecotone is ~200 m downslope from Long Lake. This means that at present, conditions are not conducive for quaking aspen to dominate at Long Lake. However, in the 1950s and ~4000 cal yr BP, we see a brief increase in quaking aspen, likely in response to the 1930/1950s drought and the 4.2ka drought. The relationship between drought, increased temperatures and widespread quaking aspen mortality has been observed across western North American, thus we hypothesize that because of drought conditions, quaking aspen (which makes up the lower treeline in the Medicine Bow Range) migrated upslope to cooler conditions. We briefly discuss both of these additional points on Page 2 in the Introduction section.

**Response to Reviewer #4**

Dear Reviewer #3. Thank you for stepping in the editorial process of this article. We feel that your comments and suggestions greatly improved the direction of the new revised manuscript. Specifically, we now explicitly state the degrees of freedom used in the Student t-test in the Methods section (see page 5 lines 27-29). In addition, we now state in plain language, the objective of the approach in the last paragraph of the Introduction (see page 4 line 10), as well as have added a new discussion section '4.3. Limitations of the modern climate analogue technique' (see page 11) where we discuss the largest assumption in our paper; the assumption that our two analogues represent megadrought conditions that occur on decadal-to-centennial timescales. In this new discussion section, we offer words of encouragement to the modelling community to further explore the scenario presented in this study.

Comment #1: the column "Purpose of Climate Variable" in Table 1 is appallingly trivial.

Response to comment #1: Thank you for pointing this out. Since we state in the results section, the objective of each climate variable, we have removed Table 1 from the manuscript.

Comment #2: There is also a mistake in the numbering of sections and subsections starting at subsection 4.1

Response to comment #2: We apologize about this mistake. We have fixed the numbering of all sections and subsections.

Comment #3: p.12 l. 33: The citation 'Using a complex numerical weather-prediction model with data from May 1987 to May 1988. Palmer and Brankovic (1989) had significant skill in prediction an anomalous high pressure ridge over North American during the summer of 1988" is about a trained prediction model, and is irrelevant in the present context.

Response to comment #3: We have deleted this sentence from the manuscript.

Comment #4: Citation "By applying the modern climate analogue technique to paleoenvironental proxies from the mid-continent, Shinker et al. (2006) found that regional moisture influx and small-scale vertical motions in the atmosphere (i.e. subsidence or uplift) provide better information regarding precipitation than large-scale general circulation alone." The wording is inaccurate. Shinker examined modern climatological data, to indeed observe better predictor of precipitation than large-scale circulation in the modern climate. They then used this insight to help and contribute to the interpretation of paleoclimate data. It is only at the latter stage that the hypothesis of an analogy applies.

Response to comment #4: We have reworded this sentence, which now states 'Shinker et al. (2006) also examined the mid-Holocene drought, but focused on the mid-continent of North America to provide potential climate processes and mechanisms associated with low lake levels during the prolonged mid-Holocene drought. They found that regional moisture influx and small-scale vertical motions in the atmosphere help explain low lake levels during that time.'

**Response to Reviewer #5**

Dear Reviewer #5. Thank you for your kind words of support. We are pleased to hear you enjoyed the paper.

Comment #1: This paper relies heavily on the pollen reconstruction in Carter et al 2013. I realized this data has already been published, but it seems very awkward that the pollen in not shown in a figure, as it is a major part of the manuscript. Perhaps the authors think this is redundant (as it is already published elsewhere), however to the reader (especially in the paleo community) it would make visualizing the changes at Long Lake through time much easier. I would encourage the authors to consider including the relevant pollen data from Long Lake either incorporated into an existing figure, or as a new figure itself.

Response to comment #1: Thank you very much for your suggestion. You are absolutely correct in that it would help the reader better visualize the changes seen at Long Lake. We now include the necessary pollen data, as well as have included the climate reconstruction from Carter et al. (2017a) to help the reader visualize the response of aspen to the megadrought identified ~4200 cal yr BP. This data is now in Figure 1 (see below). In addition, we have included a larger spatial area in this version of the manuscript to include several regional dune fields that reactivated around this time period. This paleo data is also included in the new Figure 1.

Comment #2: The authors should include the reference "Oxygen isotope records of Holocene climate variability in the Pacific Northwest" (Steinman et al., 2016) reference in line 30 on page 8 and perhaps in some other section in the paper. Steinman et al 2016 presents high-res oxygen isotopes records from three lakes that suggests the middle Holocene winters were wetter in western North America. This paper also provides model-based evidence and a robust review of paleoclimate literature from the Pacific Northwest. This may be of use to the authors during revision.

Response to comment #2: Thank you for pointing out this paper. We have included it alongside the other references on page 8 that indicate cool and wet climatic conditions during this time.

Comment #3: Lines 14-15, page 9: Steinman et al 2016 suggest a shift to more ENSO-like condition (i.e. drier winters) in the Pacific Northwest from approximately 3000-2000 cal yr BP. The references and discussion in this section needs to be updated to include other regional records.

Response to comment #3: 1. Thank you for the suggestion. However, we did not discuss this point in the manuscript for two reasons; 1) the time period 3000-2000 is beyond the inflection point of the 4.2 ka event; and 2) the PNW and its teleconnection to ENSO are very different than the CRM and Great Plains. We did include the Steinman et al. (2016) citation on page 10 line 10 as we discuss how the different regional precipitation anomalies could be the result of proxies more sensitive to winter-precipitation i.e. ENSO teleconnections.

Comment #4: I believe this was added to satisfy a reviewer, but why is the principle of uniformitarianism even stated. I feel that this is understood. I would defer to the authors discretion, but this is also awkward wording. I think something like "assuming similar conditions as modern" or something comparable would suffice.

Response to Comment #4: We have removed all discussion regarding the principle of uniformitarianism from the manuscript.

Comment #5: Lines 6-9, page 3: Two uses of the word 'analogue' in the sentence. Confusing wording.

Response to Comment #5: This sentence is describing what the modern climate analogue technique is. I believe because we forgot to include the word 'technique' after the first analogue, the reader was unable to determine the context of the sentence. We have reworded this sentence in the hopes that it is now more easily understandable; "The modern climate analogue technique is a conceptual model that uses modern extremes (e.g. drought) as analogues of past events (e.g. vegetation disturbance associated with drought in the sedimentary record) as a means to understand palaeoclimate patterns that may have caused historic paleoecological variability (Diaz and Andrews, 1982; Ely, 1997; Edwards et al., 2001; Shinker, 2014)."

Comment #6: There is a typo at the end of Line 8 on page 2.

Response to Comment #5: Thank you for pointing out this mistake. However, we have removed this paragraph from the new version.

Proposed Figure 1.

[revised manuscript text omitted]

---

## Editor Decision (ED2)

Editor comment 2 on cp-2017-107

Dear Carter et al,

Thank you for your responses to the first round of reviewers and editorial comments. We believe that the manuscript has improved.

However, the second round of reviews raises the same problems with the analogue technique again, in line with the first round of reviewers' comments (see Referee #3 additional comment 1, Referee #4 full report). This suggests that you have not addressed this issue adequately in your responses to the first round of reviews. The editorial team has therefore taken the decision to ask for a final round of major revisions, to allow you to comprehensively address this issue. If this issue cannot be addressed in this round of revisions, we will not send the paper for an additional round of revision-review and will be unable to accept the paper for publication.

In order to address the analogue technique issue, we request a paragraph in the introduction and a subsection in the discussion which clearly address the potential weaknesses and problems in application of the technique. An acceptance that this suggests more uncertainty in your results and interpretation than is currently expressed is needed. You should also discuss what might make the results presented here more robust to the challenges to the modern analogue technique and its application in this study - see comments from Referee #4 on this.

Additional comments made by referees 3, 4 and 5 should also be addressed.

We look forward to your responses and final revisions of this manuscript.

Best wishes,

Heather Plumpton (Handling Editor) and Mike Evans (Co-Editor)

---

## Author Response (AR3)

Dear Dr. Plumpton,

We would like to thank you for the wonderful news of acceptance. While the process was long, it was extremely helpful. We sincerely appreciate all the suggestions by all five referees who greatly improved the quality of the research.

We have removed the sentence is question from figures 7, 8, and 10. We also rewrote and downscaled the figure caption from Figure 1. Lastly, we did a final proof read and deleted a few typos that were found.

Please let us know if you need anything else.

Kind regards,

Dr. Vachel A. Carter